# BRINGING NERFS TO THE LATENT SPACE: INVERSE GRAPHICS AUTOENCODER

**Antoine Schnepf**[* 1,2]**, Karim Kassab**[* 1,3]**, Jean-Yves Franceschi**[1]**, Laurent Caraffa**[3]**,
**Flavian Vasile**[1]**, Jeremie Mary**[1]**, Andrew Comport**[† 2]**, Valérie Gouet-Brunet**[† 3]

[* †] Equal contribution
[1] Criteo AI Lab, Paris, France
[2] Université Côte d'Azur, CNRS, I3S, France
[3] LASTIG, Université Gustave Eiffel, IGN-ENSG, F-94160 Saint-Mandé

## ABSTRACT

While pre-trained image autoencoders are increasingly utilized in computer vision, the application of inverse graphics in 2D latent spaces has been under-explored. Yet, besides reducing the training and rendering complexity, applying inverse graphics in the latent space enables a valuable interoperability with other latent-based 2D methods. The major challenge is that inverse graphics cannot be directly applied to such image latent spaces because they lack an underlying 3D geometry. In this paper, we propose an Inverse Graphics Autoencoder (IG-AE) that specifically addresses this issue. To this end, we regularize an image autoencoder with 3D-geometry by aligning its latent space with jointly trained latent 3D scenes. We utilize the trained IG-AE to bring NeRFs to the latent space with a latent NeRF training pipeline, which we implement in an open-source extension of the Nerfstudio framework, thereby unlocking latent scene learning for its supported methods. We experimentally confirm that Latent NeRFs trained with IG-AE present an improved quality compared to a standard autoencoder, all while exhibiting training and rendering accelerations with respect to NeRFs trained in the image space. Our project page can be found at `https://ig-ae.github.io`.

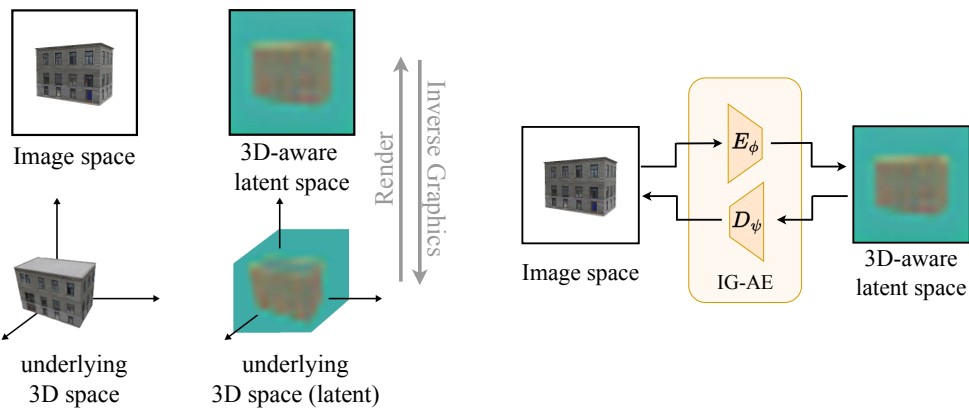

(a) Image space analogy.       (b) Inverse Graphics Autoencoder (IG-AE).

Figure 1: **3D-aware latent space.** We draw inspiration from the relationship between the 3D space and image space and introduce the concept of a 3D-aware latent space. We propose an Inverse Graphics Autoencoder (IG-AE) that encodes images into 3D-aware latent images, hence preserving 3D-consistency. We use these latents to train scene representations in the 3D-aware latent space.

# 1 INTRODUCTION

Latent image representations are increasingly used in computer vision tasks. Most recent methods utilize latent representations for various tasks such as image generation (Rombach et al., 2022; Esser et al., 2021), and image segmentation (Long et al., 2015; Ohgushi et al., 2020), as they allow to represent images in a compact form which reduces their representation complexity.

Similarly to 2D computer vision, leveraging latent image representations for 3D tasks would bring numerous advantages. First, training NeRFs on lower-resolution latent images would enable significant speed-ups in both rendering and training times. While efforts towards NeRF speed improvements have traditionally focused on working directly on NeRF architectures, this approach would instead tackle the space in which NeRF models are trained. This means that it would offer a broader impact across various NeRF models, extending beyond a single specific NeRF improvement. Second, latent NeRFs unlock an inter-operability between NeRFs and pre-trained latent-based methods. For instance, previous works have enabled this inter-operability in a task-specific, ad-hoc manner to unlock applications like scene editing (Park et al., 2024) and generation (Metzer et al., 2023).

Despite the potential of latent NeRFs, the adoption of latent image representations for 3D tasks, such as Novel View Synthesis (NVS), remains limited due to the inherent incompatibilities between image latent spaces and 3D modeling methods. These incompatibilities stem from the lack of an underlying 3D geometry in latent spaces on the one hand, and the necessity of 3D-consistency among images for solving the inverse graphics problem on the other hand. In practice, encoding two 3D-consistent views of an object does not lead to 3D-consistent latent representations; instead, it produces latent images with features that vary inconsistently across views. This prohibits the direct application of scene learning methods such as NeRF (Mildenhall et al., 2020) on such representations.

To sidestep these incompatibilities, the aforementioned methods have resorted to the ad-hoc implementation of custom, scene-dependent "adapters" (Khalid et al., 2023) or "refinement layers" (Park et al., 2024). However, these solutions are only workarounds and do not address the root cause of the problem: the need for a universal "3D-aware" latent space induced by an image autoencoder that preserves 3D-consistency. Such an autoencoder would enable a broader compatibility between image autoencoders and NeRFs, unlocking the potential of latent NeRFs. In this paper, we explore this direction for the first time.

We start by proposing a technique that enables learning NeRF models in the latent space. Our training pipeline comprises two stages: **Latent Supervision** supervises NeRFs with latent image representations, and **RGB Alignment** aligns the learned latent scene with the RGB space. Yet, its application in a standard latent space results in sub-optimal latent NeRFs, due to their aforementioned incompatibilities. To address this, we draw inspiration from the relationship between the image space and 3D space (Fig. 1a) and introduce 3D-aware latent spaces, which augment regular image latent spaces by incorporating an underlying 3D geometry. We present an **Inverse Graphics Autoencoder** (IG-AE) that embeds a **3D-aware latent space** (Fig. 1b). To achieve this, we apply 3D regularization on the latent space of an autoencoder by jointly training it with a synthetic set of latent 3D scenes. During this process, we align the encoded views with the renderings of the latent 3D scenes, which enforces 3D-consistency. Additionally, we propose an autoencoder preservation process that simultaneously autoencodes real and synthetic data while performing our training, which allows us to conserve the reconstructive performance of the autoencoder.

We propose an open-source extension of Nerfstudio (Tancik et al., 2023) that enables learning its supported NeRF models in latent spaces, thereby unlocking a streamlined approach for latent NeRF learning. This enables us to evaluate various current NeRF architectures on the task of learning scenes in latent spaces. Our results highlight the effectiveness of our IG-AE in latent scene learning as compared to a standard AE. We consider this work to be the first milestone towards foundation inverse graphics autoencoders, and aspire that our open-source Nerfstudio extension promotes further research in this direction.

An overview of our contributions is given below.

- We introduce the concept of 3D-aware latent spaces that are compatible with 3D tasks,
- We present an Inverse Graphics Autoencoder (IG-AE) that maps images to a 3D-aware latent space, all while preserving autoencoder performances,

- We propose a standardized method to train NeRF architectures in latent spaces,

- We propose an open-source extension of Nerfstudio that support training supported NeRF models in the latent space, in an effort to streamline future work involving latent NeRFs.

- We experimentally show that IG-AE enables improved latent NeRF quality with respect to the baseline AE and decreased training times with respect to regular NeRFs.

## 2 RELATED WORK

**Latent NeRFs.**  While NeRFs (Mildenhall et al., 2020) were originally conceived to work in the image space, they have also been extended to work in other feature spaces (Tschernezki et al., 2022; Kobayashi et al., 2022). Notably, Latent NeRFs are extensions of NeRFs to the latent spaces of image autoencoders. They have been particularly explored in works targeting applications such as scene editing (Park et al., 2024; Khalid et al., 2023) and text-to-3D generation (Metzer et al., 2023), which are not directly feasible with regular NeRFs. To circumvent the incompatibilities between NeRFs and latent spaces, these works have resorted to special adapter layers that correct NeRF renderings into standard latent image representations. For novel view synthesis, Aumentado-Armstrong et al. (2023) employ hybrid NeRFs that are trained to simultaneously render both RGB and latent components. This is done to supervise the NeRF geometry during training, while only keeping latent components at inference, which enables good NVS performances and rendering speed-ups. In this work, we aim to train NeRFs that operate fully within the latent space. We address the incompatibility between NeRFs and latent spaces by tackling its root cause: the need for an inverse graphics autoencoder that preserves 3D consistency, or in other words, the need for a 3D-aware latent space.

**Scene embeddings.**  While methods like Latent NeRFs and other feature fields modify the NeRF rendering space by replacing scene images with "feature images", other approaches embed the entire scene information into a "scene embedding". Recent works (Wang et al., 2023; Kosiorek et al., 2021; Sajjadi et al., 2022; Lan et al., 2024) focus on training encoders to transform scene information (e.g. images or text) into embeddings that fully encapsulate the 3D scene. NeRFs can then be obtained directly (without training) from such scene embeddings. In contrast, our approach aligns more closely with feature field methods (Tschernezki et al., 2022; Kobayashi et al., 2022), where NeRFs are trained to render "feature images" derived from scene images (e.g. DINO or CLIP features), enabling applications such as segmentation and object retrieval. However, we train NeRFs to render latent image representations obtained from an autoencoder fine-tuned for 3D tasks, to tackle the task of 3D reconstruction. For an elaborate distinction across recent works, we refer the reader to Appendix A.

**Nerfstudio.**  Subsequent to the introduction of NeRF (Mildenhall et al., 2020), a multitude of NeRF models emerged to improve upon the original architecture by accelerating training times (Müller et al., 2022; Kerbl et al., 2023), improving rendering quality (Barron et al., 2021; Chen et al., 2022), as well as targeting other limitations (Fridovich-Keil et al., 2023; Martin-Brualla et al., 2021; Yu et al., 2021). With the escalation of NeRF methods, Nerfstudio (Tancik et al., 2023) emerged as a unified PyTorch (Paszke et al., 2019) framework in which NeRF models are implemented using standardized implementations, making it straightforward for researchers and practitioners to integrate various NeRF models into their projects. Today, most well-known NeRF techniques are implemented in Nerfstudio (Müller et al., 2022; Barron et al., 2021; Mildenhall et al., 2020; Kerbl et al., 2023), thus allowing for a friction-free experience when testing and comparing novel techniques. While Nerfstudio currently provides a streamlined approach for NeRFs, it limits the representation of 3D scenes to the natural color space, hence limiting both the research and development of Latent NeRFs. In this work, we propose an open-source extension of the Nerfstudio framework that enables training supported NeRF methods in a custom latent space, thus facilitating our current as well as future research in this area. Furthermore, we integrate our IG-AE into this extension, which enables us to evaluate various current NeRF architectures on the task of latent scene learning.

## 3 LATENT NERF

In this section, we start by presenting latent NeRFs. Then, we propose a general latent NeRF training method, which will later serve to train NeRFs in the latent space of our IG-AE. Provided an encoder

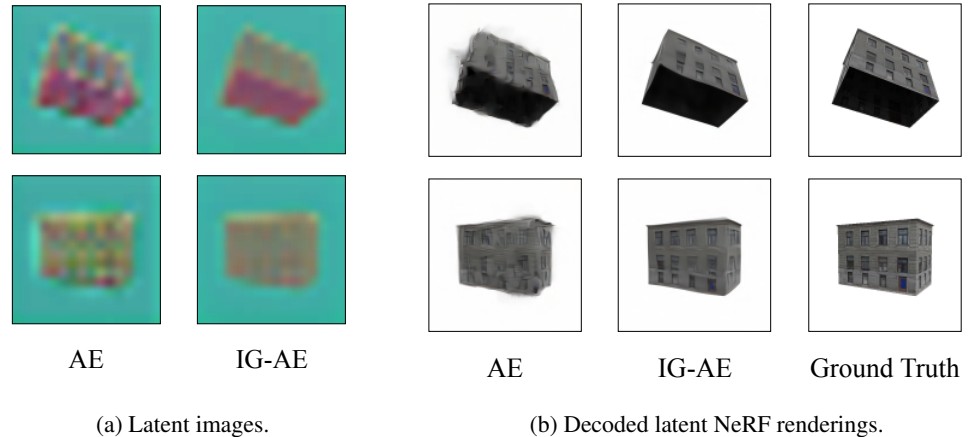

|          |          |          |          |          |
|:--------:|:--------:|:--------:|:--------:|:--------:|
| AE | IG-AE | AE | IG-AE | Ground Truth |
| (a) Latent images. | | (b) Decoded latent NeRF renderings. | | |

Figure 2: **Comparison of IG-AE and a standard AE.** Encoding 3D-consistent images using an AE leads to 3D-inconsistent latent images. When trained on such latents, NeRF renderings present artifacts when decoded. IG-AE presents a 3D-aware latent space with 3D-consistent latent images. Latent NeRFs trained with IG-AE eliminate these artifacts and more closely match the ground truth.

$E_\phi$ and decoder $D_\psi$, our method can train any NeRF architecture in a latent space via two stages. **Latent Supervision** is an adaptation of the standard NeRF training framework to the latent space which replaces RGB images with latent images. **RGB Alignment** fine-tunes the decoder to align the learned latent scene with the RGB space, which our experiments proved to be essential for strong NVS performances in the RGB space.

## 3.1 Definition

Latent NeRFs are conceptually similar to standard NeRFs, with the primary difference being that they model scenes in the latent space of an autoencoder as opposed to the RGB space. As such, latent NeRFs are simple extensions of standard NeRF methods, where the rendering resolution and the number of output channels are modified in accordance with the latent space dimensions. Let $F_\theta$ be a latent NeRF method with trainable parameters $\theta$, where $F_\theta \in \{$Vanilla-NeRF (Mildenhall et al., 2020), Instant-NGP (Müller et al., 2022), TensoRF (Chen et al., 2022), K-Planes (Fridovich-Keil et al., 2023)$\}$. For conciseness, we consider $F_\theta$ to be a generic NeRF method and abstract from method-specific nuances. Given a camera pose $p$, one can render a novel view as such:

$$\tilde{z}_p = F_\theta(p) \,, \qquad\qquad \tilde{x}_p = D_\psi(\tilde{z}_p) \,, \qquad\qquad (1)$$

where $\tilde{z}_p$ is the rendered latent image of shape $(h, w, c)$, and $\tilde{x}_p$ is the decoded RGB image of shape $(H, W, 3)$. We define $l > 1$ as the resolution downscale factor when going from the RGB space to the latent space: $(h, w) = (H/l, W/l)$. As NeRF rendering pipelines implement classic volume rendering (Kajiya & Von Herzen, 1984) where pixels are independently rendered, latent NeRFs reduce the rendering complexity by a factor of $l^2$ as compared to standard NeRFs. This makes latent NeRFs highly attractive, as they alleviate a key bottleneck in NeRF training, while maintaining the same target resolution after decoding.

## 3.2 Training

Our latent NeRF training scheme consists of two stages, as illustrated in Fig. 3.

**Latent Supervision.** In this stage, we consider the direct adaptation of NeRF training in the latent space, i.e. training NeRF on latent image representations rather than RGB images. Let $\mathcal{L}_{F_\theta}$ be the training loss corresponding to the NeRF method $F_\theta$. $\mathcal{L}_{F_\theta}$ comprises of a pixel-level reconstructive objective that matches the NeRF renderings to ground truth views, as well as other method-specific

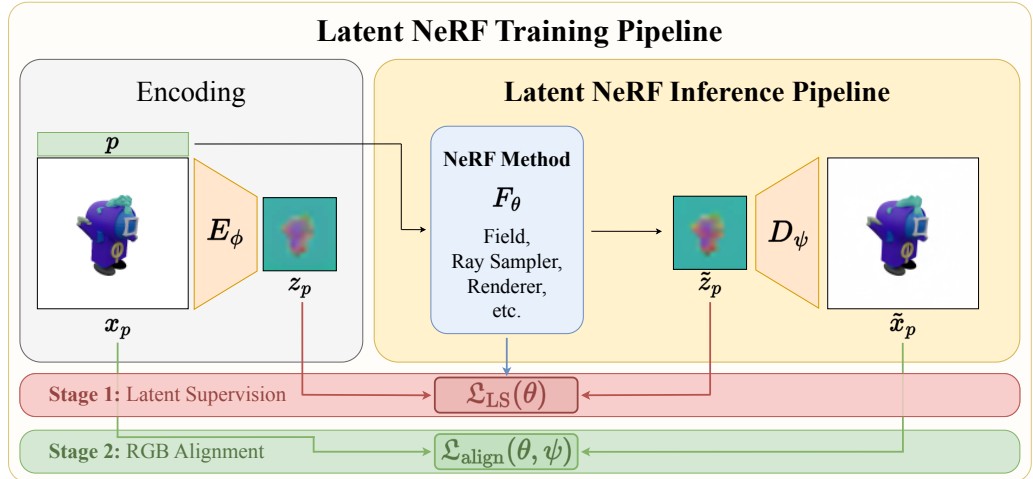

Figure 3: **Latent NeRF Training.** We train a Latent NeRF in two stages. First, we train the chosen NeRF method $F_\theta$ on posed encoded latent images using its proprietary loss $\mathcal{L}_{F_\theta}$ that matches rendered latents $\tilde{z}_p$ and encoded latents $z_p$. Subsequently, we align with the scene in the RGB space by adding decoder fine-tuning via $\mathcal{L}_{\text{align}}$ that matches ground truth images $x_p$ and decoded renderings $\tilde{x}_p$.

regularization terms. Latent Supervision consists of minimizing the following objective:

$$\mathfrak{L}_{\text{LS}}(\theta) = \sum_{p \in \mathcal{P}} \mathcal{L}_{F_\theta}(\theta; z_p, \tilde{z}_p) \, , \tag{2}$$

where $z_p$ and $\tilde{z}_p$ are respectively the encoded latent representation of the RGB ground truth and the rendered latent image, with pose $p$. $\mathcal{P}$ denotes the set of training camera poses. Note that, to avoid redundantly encoding training views, we start this stage by encoding all the training images $\mathcal{X} = \{x_p\}_{p \in \mathcal{P}}$ into latent image representations $\mathcal{Z} = \{z_p\}_{p \in \mathcal{P}}$, and then caching them.

**RGB alignment.** While Latent Supervision effectively captures a 3D structure in the latent space, even minor inaccuracies in latent NeRF renderings can be magnified during the decoding process. These inaccuracies stem from various sources of error within the pipeline, mainly: errors originating from the latent space and autoencoding performances (i.e. imperfect AE), and errors associated with 3D modeling (i.e. imperfect NeRF). Hence, in order to alleviate the effect of these errors, we finish our latent NeRF training by an RGB alignment process, where we fine-tune the decoder $D_\psi$ and the latent scene $F_\theta$ to align with the RGB space. In practice, we minimize the following objective:

$$\mathfrak{L}_{\text{align}}(\theta, \psi) = \sum_{p \in \mathcal{P}} \|x_p - \tilde{x}_p\|_2^2 \, , \tag{3}$$

where $x_p$ is the RGB ground truth, and $\tilde{x}_p = D_\psi(\tilde{z}_p)$ is the decoded latent NeRF rendering. Consequently, the latent NeRF not only exhibits good NVS performances in the latent space, but also when decoding its renderings to the RGB space.

**Overall,** we divide latent NeRF training into Latent Supervision and RGB Alignment. Latent Supervision is an accelerated approach that learns 3D structures in the latent space thanks to reduced image resolutions, and RGB alignment enhances NVS quality in the RGB space by using RGB supervision and a decoder fine-tuning, while still rendering the NeRF in the latent space. Fig. 2 (AE columns) illustrates the results of our latent NeRF training method when applied in a standard latent space. Latent NeRFs trained in a standard latent space with our method learn a coarse geometry. However, due to the aforementioned incompatibilities between latent spaces and NeRFs, the decoded renderings present artifacts in the RGB space. To address this, we shift our focus to resolving these incompatibilities in the next section, and present an inverse graphics autoencoder embedding a 3D-aware latent that is compatible with learning latent NeRFs.

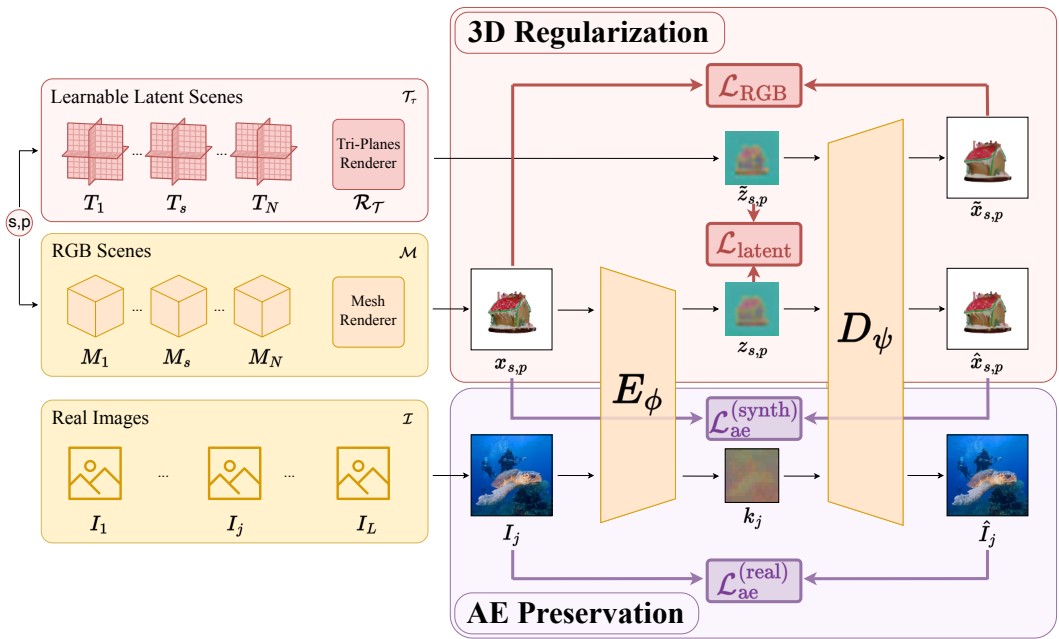

Figure 4: **IG-AE Training.** We jointly learn a set of latent synthetic scenes $\mathcal{T}_\tau$ and supervise the latent images $z_{s,p}$ of an autoencoder with rendered 3D-consistent latents $\tilde{z}_{s,p}$ using $\mathcal{L}_{\text{latent}}$. We match decoded latent renderings $\tilde{x}_{s,p}$ with the ground truth scene renderings $x_{s,p}$ using $\mathcal{L}_{\text{RGB}}$. We preserve autoencoder performances on synthetic and real data respectively through $\mathcal{L}_{\text{ae}}^{(\text{synth})}$ and $\mathcal{L}_{\text{ae}}^{(\text{real})}$.

## 4    INVERSE GRAPHICS AUTOENCODER

We present IG-AE, an autoencoder that encodes 3D-consistent images into 3D-consistent latent representations. To attain such an autoencoder, it is necessary that its latent space encodes the RGB space while also retaining an underlying 3D geometry. In this section, we start by defining 3D-consistency (Section 4.1), and then elaborate on how we train an IG-AE: we utilize synthetic data to construct a learnable set of latent scenes which aims to supervise the latent space of an AE with 3D-consistent latents (Section 4.2), all while preserving autoencoding performances (Section 4.3). Fig. 4 presents an overview of our method.

### 4.1    3D-CONSISTENCY

The notion of 3D consistency ensures that corresponding points or features in different images represent the same point or object in the scene, despite variations in viewpoint, lighting or occlusion. We denote the posed images as $\mathcal{X} = \{x_p \mid p \in \text{SE}(3)\}$, where $\text{SE}(3)$ is the Special Euclidean group containing all camera poses in 3D. $\mathcal{X}$ are posed 3D-consistent images if and only if there exists a 3D model $M$ such that:

$$\forall p \in \text{SE}(3), \ \text{Render}(M, p) = x_p . \tag{4}$$

Note that a NeRF model inherently produces 3D-consistent images as it is an implicit model $M$ rendered via classic volume rendering (Kajiya & Von Herzen, 1984). Also note that while 3D consistency is natural for posed images $\mathcal{X} = \{x_p\}_{p \in \mathcal{P}}$ obtained from a scene in the image space, it does not naturally extend to the latent space, as latent representations of two 3D-consistent images are not necessarily 3D consistent. Our 3D-aware latent space is designed to mitigate this discrepancy.

### 4.2    3D-REGULARIZATION

To achieve 3D-consistency in the latent space, we learn an IG-AE by aligning the latent encodings of 3D-consistent images with reference 3D-consistent latent images. However, this cannot be directly achieved as such 3D-consistent latent images are not available. To this end, we learn a set of 3D latent

scenes with NeRFs from which 3D-consistent latents can be easily rendered. In fact, while NeRFs cannot replicate 3D-inconsistent latents from a given autoencoder, training them via $\mathcal{L}_{F_\theta}$ (which typically employs an MSE reconstructive objective) leads to a convergence towards a common coarse geometry that most satisfies the inputs, while maintaining 3D-consistency. We utilize this property in our approach to obtain 3D-consistent latent images with which we supervise our autoencoder. To learn our latent scenes in practice, we adopt Tri-Plane representations (Chan et al., 2022), as their simple architecture ensures a low memory footprint and a fast training.

Let $\mathcal{M} = \{M_1, ..., M_N\}$ be a dataset of 3D scenes, and $\mathcal{X}_s = \{x_{s,p}\}_{p \in \mathcal{P}}$ be the set of renderings of a scene $M_s \in \mathcal{M}$ from an array of training views $\mathcal{P}$. We denote the set of latent scenes as the set of Tri-Planes $\mathcal{T}_\tau = \{T_1, ..., T_N\}$, where $\tau$ comprises both scene-specific trainable parameters, as well as shared parameters present in the feature decoder of our common Tri-Plane renderer $\mathcal{R}_\mathcal{T}$.

Before training our IG-AE, we start by encoding views of our scenes $\mathcal{M}$ into standard (i.e. 3D-inconsistent) latent views, and training our Tri-Planes on these latents. Subsequently, we proceed to train our IG-AE while continuing to jointly train our Tri-Planes. For each scene $M_s$, we encode each view $x_{s,p}$ into a latent image $z_{s,p}$, and render the corresponding 3D-aware latent image $\tilde{z}_{s,p}$ from the latent scene $T_s$ as follows:

$$z_{s,p} = E_\phi(x_{s,p}) , \qquad\qquad \tilde{z}_{s,p} = \mathcal{R}_\mathcal{T}(T_s, p) , \qquad\qquad (5)$$

where $E_\phi$ is the encoder with trainable parameters $\phi$. We then define a loss function $\mathcal{L}_{\text{latent}}$ that aligns these latent representations:

$$\mathcal{L}_{\text{latent}}(\phi, \tau; z_{s,p}, \tilde{z}_{s,p}) = \|z_{s,p} - \tilde{z}_{s,p}\|_2^2 . \qquad\qquad (6)$$

This loss function updates both the encoder parameters $\phi$ and the latent scene parameters $\tau$ to minimize the distance between the latent representations. This means that, on the one hand, the latent scene $T_s$ is updated to align with the 3D-inconsistent latent images $z_{s,p}$, and hence learn a coarse geometry common among all latent images of $M_s$. On the other hand, the encoder parameters $\phi$ are updated to produce latent images that more closely reassemble the latent scene renderings $\tilde{z}_{s,p}$, or in other words, produce 3D-consistent images.

We additionally define $\mathcal{L}_{\text{RGB}}$ as a loss function that mirrors $\mathcal{L}_{\text{latent}}$ in the RGB space:

$$\mathcal{L}_{\text{RGB}}(\psi, \tau; x_{s,p}, \tilde{x}_{s,p}) = \|x_{s,p} - \tilde{x}_{s,p}\|_2^2 , \qquad\qquad (7)$$

where $\tilde{x}_{s,p} = D_\psi(\tilde{z}_{s,p})$ is the decoded latent scene rendering. $\mathcal{L}_{\text{RGB}}$ updates both the decoder parameters $\psi$ and the latent scene parameters $\tau$ so that the latent scene aligns with the RGB scene when decoded. This is to ensure the optimal decoding of 3D-consistent latents as well as NVS performances in the RGB space.

Therefore, our 3D regularization objective minimizes the following loss:

$$\mathfrak{L}_{3\text{D}}(\phi, \psi, \tau) = \sum_{s,p} \left[ \lambda_{\text{latent}} \mathcal{L}_{\text{latent}}(\phi, \tau; z_{s,p}, \tilde{z}_{s,p}) + \lambda_{\text{RGB}} \mathcal{L}_{\text{RGB}}(\psi, \tau; x_{s,p}, \tilde{x}_{s,p}) \right] , \qquad (8)$$

where $\lambda_{\text{latent}}$ and $\lambda_{\text{RGB}}$ are hyper-parameters.

**Overall,** our IG-AE is trained by aligning the latent space of an autoencoder with the space of latent images that is inferable via Tri-Planes, while also ensuring the proper mapping from the latent space to the RGB space. In practice, we use synthetic scenes from Objaverse (Deitke et al., 2023) to learn 3D structure in the latent space, as they present well-defined geometry and error-free camera parameters. Fig. 2 illustrates a comparison between a standard AE latent space and our 3D-aware latent space. While latent NeRFs trained with an AE present artifacts when decoding their renderings, those trained in the latent space of our IG-AE do not exhibit such artifacts. Yet, our experiments show that solely applying our 3D-regularization optimization objective leads to a degradation of the AE's generalization capabilities and autoencoding performances. On that account, we present in the next section additional components of our training that target a reconstructive objective.

## 4.3 AUTOENCODER PRESERVATION

As 3D regularization does not incorporate autoencoder reconstruction, this section focuses on preserving the autoencoding performance of the AE on both synthetic and real data. To this end, we

additionally jointly learn a reconstructive objective. First, we add an autoencoder loss to align the ground truth RGB views with the reconstructed views after encoding and decoding:

$$\mathcal{L}_{\text{ae}}^{(\text{synth})}(\phi, \psi; x_{s,p}, \hat{x}_{s,p}) = \|x_{s,p} - \hat{x}_{s,p}\|_2^2 \, , \tag{9}$$

where $\hat{x}_{s,p} = D_\psi(z_{s,p})$ is the reconstruction of the ground truth image. This ensures that the autoencoder still reconstructs images from the scenes.

To avoid overfitting on synthetic data, we additionally inject real images from Imagenet (Deng et al., 2009) into our training pipeline, on which we also define a reconstructive objective. We denote $\mathcal{I} = \{I_1, ..., I_L\}$ the dataset of real images. We define the reconstructive loss on an image $I_j \in \mathcal{I}$ as:

$$\mathcal{L}_{\text{ae}}^{(\text{real})}(\phi, \psi; I_j, \hat{I}_j) = \|I_j - \hat{I}_j\|_2^2 + \lambda_{\text{p}}\mathcal{L}_{\text{p}}(I_j, \hat{I}_j) + \lambda_{\text{TV}}\mathcal{L}_{\text{TV}}(k_j) \, , \tag{10}$$

where $\lambda_{\text{p}}$ and $\lambda_{\text{TV}}$ are hyper-parameters, $\hat{I}_j = D_\psi(k_j)$ is the reconstruction of $I_j$, and $k_j = E_\phi(I_j)$ is the encoding of $I_j$. $\mathcal{L}_{\text{p}}$ is a perceptual loss that refines the autoencoder reconstructions by aligning the reconstructed and original images in the feature space of a pre-trained network. $\mathcal{L}_{\text{TV}}$ is a total variation loss that acts as a regularization term to encourage spatial smoothness by penalizing high-frequency variations in the latent images. We found that it leads to latent images that more closely reassemble our 3D-consistent latents. This is done to prevent the IG-AE from converging towards a dual-mode solution, where it is only 3D-consistent for synthetic scenes while functioning as a normal AE for real scenes. More details about TV are present in Appendix C.

Therefore, our autoencoder preservation loss is defined as follows:

$$\mathfrak{L}_{\text{ae}}(\phi, \psi) = \lambda_{\text{ae}}^{(\text{synth})} \sum_{s,p} \mathcal{L}_{\text{ae}}^{(\text{synth})}(\phi, \psi; x_{s,p}, \hat{x}_{s,p}) + \lambda_{\text{ae}}^{(\text{real})} \sum_{j} \mathcal{L}_{\text{ae}}^{(\text{real})}(\phi, \psi; I_j, \hat{I}_j) \, , \tag{11}$$

where $\lambda_{\text{ae}}^{(\text{synth})}$ and $\lambda_{\text{ae}}^{(\text{real})}$ are hyper-parameters.

## 4.4 IG-AE TRAINING OBJECTIVE

Overall, we present a joint objective for IG-AE that regularizes its latent space with 3D geometry while preserving autoencoding performances. The objective minimizes the following loss:

$$\mathfrak{L}_{\text{IG-AE}}(\phi, \psi, \tau) = \mathfrak{L}_{\text{3D}}(\phi, \psi, \tau) + \mathfrak{L}_{\text{ae}}(\phi, \psi) \, . \tag{12}$$

In summary, our training strategy exploits the well-defined geometry of synthetic data and utilizes $\mathfrak{L}_{\text{3D}}$ to learn latent scenes and regularize the latent space of an AE with 3D geometry, all while maintaining the reconstruction performance of the AE via $\mathfrak{L}_{\text{ae}}$ and real data.

## 5 EXPERIMENTS

We start by training an IG-AE on synthetic scenes from Objaverse, as described in Section 4. Subsequently, we train Nerfstudio models in our 3D-aware latent space on scenes from *out-of-distribution* datasets, as well as held-out scenes from Objaverse. To evaluate our method, we compare the novel view synthesis performance of the Nerfstudio models when trained in our 3D-aware latent space (IG-AE), a standard latent space (AE), and the RGB space. Moreover, we evaluate the auto-encoding performances of our IG-AE on held-out real images. Finally, we assess our design choices by presenting an ablation study of our method, where we omit either our 3D-regularization components or our AE preservation components. The details of our Nerfstudio extension, as well as IG-AE and latent NeRF trainings, are available in Appendix B.

## 5.1 DATASETS

**For 3D-regularization,** we adopt Objaverse (Deitke et al., 2023), a synthetic dataset which is standard when large-scale and diverse 3D data is needed (Liu et al., 2023; Shi et al., 2024).

Objaverse provides meshes of synthetic objects, and thus presents no artifact in its scenes and no approximation errors in camera parameters. We utilize $N = 500$ objects from Objaverse. Each object is rendered from $V = 300$ views at a $128 \times 128$ resolution.

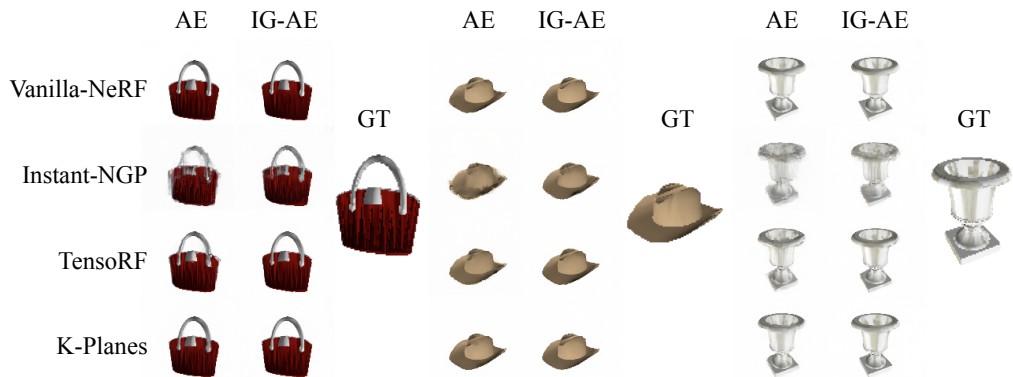

Figure 5: **Qualitative results.** Visualization of decoded latent NeRF renderings trained with a standard AE and an IG-AE on scenes from three *out-of-distribution* datasets. Latent NeRFs trained with an AE exhibit artifacts in decoded renderings that are not present in those trained with IG-AE.

Table 1: **Main Results on ShapeNet datasets.** All results are obtained by training NeRFs with our Latent NeRF Training Pipeline, and are averaged over 4 scenes from each dataset. Our 3D-aware latent space generalizes to *out-of-distribution* datasets and is more suited for latent NeRF training.

| | | Bags dataset | | | Hats dataset | | | Vases dataset | | |
|---|---|---|---|---|---|---|---|---|---|---|
| | | PSNR | SSIM | LPIPS | PSNR | SSIM | LPIPS | PSNR | SSIM | LPIPS |
| Vanilla-NeRF | AE | 28.81 | 0.954 | 0.046 | 28.75 | 0.960 | **0.032** | 32.00 | 0.968 | 0.023 |
| | IG-AE | **29.57** | **0.960** | **0.044** | **29.31** | **0.967** | 0.033 | **33.06** | **0.973** | **0.021** |
| Instant-NGP | AE | 24.29 | 0.892 | 0.113 | 23.48 | 0.893 | 0.096 | 26.49 | 0.917 | 0.081 |
| | IG-AE | **25.90** | **0.923** | **0.062** | **25.30** | **0.925** | **0.048** | **28.44** | **0.942** | **0.043** |
| TensoRF | AE | 26.19 | 0.930 | 0.060 | 25.90 | 0.932 | 0.044 | 29.16 | 0.948 | 0.038 |
| | IG-AE | **28.40** | **0.953** | **0.038** | **28.09** | **0.957** | **0.029** | **31.45** | **0.966** | **0.021** |
| K-Planes | AE | 28.00 | 0.946 | 0.041 | 27.68 | 0.951 | 0.031 | 31.56 | 0.964 | 0.023 |
| | IG-AE | **29.22** | **0.957** | **0.038** | **28.81** | **0.962** | **0.027** | **32.79** | **0.971** | **0.019** |

**For AE preservation,** we adopt Imagenet (Deng et al., 2009), a large dataset of diverse real images. We utilize $L = 40\,000$ images, which we pre-process by doing a square cropping followed by a downscaling to a $128 \times 128$ resolution that matches our rendered scenes.

**For NeRF evaluations,** we utilize synthetic, object-level data as it aligns with the training domain. As such, we train NeRFs on held-out scenes from Objaverse, and on scenes from three out-of-distribution datasets: Shapenet Hats, Bags, and Vases (Chang et al., 2015). Note that, due to the challenges latent NeRFs face with simple object-level scenes, our evaluation focuses on such scenes, and excludes more complex or real scenes. We discuss this in Section 5.4.

## 5.2 RESULTS

We report for each experiment the PSNR (↑), the SSIM (↑), and the LPIPS (↓) (Zhang et al., 2018). Quantitative and qualitative results can be found in Table 1 and Fig. 5, where we utilize our latent NeRF training pipeline to train various Nerfstudio models on out-of-distribution datasets from ShapeNet, both using our IG-AE and a standard AE. Models trained with IG-AE showcase superior NVS performance compared to those trained with a standard AE. This shows that IG-AE embeds a 3D-aware latent space better suited for NeRF training and capable of generalizing to new datasets. Additional results on held-out Objaverse scenes can be found in Appendix D.3. To further support this, Table 6 presents latent NVS metrics computed on latent images. Latent NeRFs showcase significantly better latent NVS performance with IG-AE, further confirming its 3D-awareness. Table 4 presents the NVS performance in the two stage of Latent NeRF training. In Table 8, we provide a comparison of latent NeRF training with classical RGB training. Latent NeRF training showcases speedups both in terms of training and rendering times, but lower NVS performance, which we discuss in Section 5.4.

Table 2: **Ablations.** Comparison between IG-AE and the baseline AE on two tasks. First, we compare their NVS performances when used to train a latent Vanilla-NeRF. Second, we compare their reconstruction performances when auto-encoding held-out images from ImageNet. Our method presents both strong NVS and reconstruction performances as compared to its ablations, validating our design choices. The **bold** and underlined entries indicate the best and second-best results.

| | NVS | | | | | | Reconstruction | | |
| | Cake | | | Figurine | | | | | |
| | PSNR | SSIM | LPIPS | PSNR | SSIM | LPIPS | PSNR | SSIM | LPIPS |
|---|---|---|---|---|---|---|---|---|---|
| AE | 33.18 | 0.962 | 0.053 | 31.46 | 0.972 | 0.017 | 27.69 | 0.856 | 0.023 |
| IG-AE (no 3D) | 32.09 | 0.951 | 0.063 | 30.94 | 0.965 | 0.021 | **29.83** | **0.898** | **0.013** |
| IG-AE (no Pr) | 33.87 | **0.966** | **0.050** | **33.73** | **0.982** | **0.012** | 17.66 | 0.410 | 0.279 |
| IG-AE | **34.38** | **0.966** | 0.051 | 33.17 | 0.979 | **0.012** | 29.57 | 0.887 | 0.015 |

## 5.3 Ablations

To justify our design choices, we conduct an ablation study in which we alternatively omit our 3D regularization pipeline and our AE preservation components. Ablation results on held-out Objaverse scenes for Vanilla NeRF can be found in Table 2. "**IG-AE (no Pr)**" removes the AE preservation components from our training by deactivating $\mathcal{L}_{ae}$. "**IG-AE (no 3D)**" omits our 3D regularizatiom by deactivating $\mathcal{L}_{3D}$. As illustrated, IG-AE (no Pr) presents deteriorated autoencoding performance due to overfitting on the task of 3D-regularization. IG-AE (no 3D) showcases good autoencoding performances, but lower NVS performances, as the latent space here is not 3D-aware due to the lack of 3D-regularization. Our method (IG-AE) demonstrates both good NVS and autoencoding reconstruction performances as compared to its ablations, thereby justifying the components of our pipeline. Note that ablating both our 3D regularization components and our AE preservation components leads back to the case of a standard autoencoder (AE). Additional ablations done with other NeRF models can be found in Table 5 in Appendix D.2.

## 5.4 Limitation

Table 8 compares latent NeRF training with classical RGB training. Despite speedups both in terms of training and rendering times, latent NeRFs have a lower NVS performance. This is due to a limitation in the representation of high frequencies in latent NeRFs. We speculate that this issue arises because enforcing 3D consistency in the latent space tends to prioritize regularizing low-frequency structures over high-frequency details. This leads to the loss of high-frequency details when decoding latent NeRF renderings back to the RGB space, as it can be seen in Fig. 5. Furthermore, this currently hinders the applicability of latent NeRFs on complex scenes with abundant high-frequency details, such as real-world scenes. We leave this to future work, as our approach is the first to propose a 3D-aware latent space via an image autoencoder, and is compatible with various NeRF architectures.

## 6 Conclusion

In this paper, we propose IG-AE, the first image autoencoder embedding a 3D-aware latent space. Moreover, we present a latent NeRF training pipeline that brings NeRF architectures to the latent space. We integrate our pipeline into an open-source extension of Nerfstudio, thereby enabling latent NeRF training for its supported architectures. Extensive experiments show the notable improvements our 3D-aware latent space brings as compared to training NeRFs in a standard latent space, as well as the efficiency improvements it brings with respect to training NeRFs in the RGB space. In concluding this paper, several directions of future work arise, as we consider this work to be the first milestone towards foundation inverse graphics autoencoders. This includes, for instance, the exploration of approaches improving the representation of high-frequency details when decoding latent NeRFs. We hope that our proposed Nerfstudio extension, as well as our open-source codebase, will promote further research in this direction.

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

# A  RELATED WORK

Table 3: **Comparison with related work**. Our work is the first to train various NeRF architectures in the latent space of an autoencoder, where NeRFs are able to render images that match AE latent, thanks to the 3D-aware latent space of our IG-AE.

| Method | Rendering space | 3D alignment of rendering and reference spaces | NeRF Repr. | Task |
|---|---|---|---|---|
| NeRF-VAE | RGB | yes (RGB) | Vanilla-NeRF | Generation |
| Rodin | RGB | yes (RGB) | Tri-Planes | Generation |
| LN3Diff | RGB | yes (RGB) | Tri-Planes | Generation |
| StyleNeRF | Feature | — | Vanilla-NeRF | Generation |
| Latent-NeRF | VAE Latent (Train) RGB (Inference) | yes (RGB) | I-NGP | Generation |
| LERF | RGB & CLIP | yes (RGB) | Vanilla-NeRF | 3D Distillation |
| N3F | RGB & DINO | no (DINO) | Tri-Planes | 3D Distillation |
| Latent-Editor | VAE Latent | no (VAE) | Vanilla-NeRF | 3D Editing |
| ED-NeRF | VAE Latent | no (VAE) | TensoRF | 3D Editing |
| RLS | RGB & VAE Latent (Train) VAE Latent (Inference) | no (VAE) | Vanilla-NeRF | Reconstruction |
| Ours | IG-AE Latent (3D-aware) | yes (IG-AE) | Vanilla-NeRF, K-Planes, TensoRF, Instant-NGP | Reconstruction |

This section is dedicated to compare some recent works utilizing NeRFs with scene embeddings and feature images. Table 3 summarizes this section.

As discussed in the related work (Section 2), some methods encode scene information (e.g. images, text) into a scene embedding that then serves to produce a NeRF. Such works do not tackle changing the RGB rendering space of NeRFs.

- **NeRF-VAE** (Kosiorek et al., 2021) trains a VAE to encode scene images into a scene embedding. This embedding is then used to condition a NeRF model. They can additionally sample such embeddings in the VAE latent space, which allows for scene generation.
- **Rodin** (Wang et al., 2023) enables scene generation with an optional conditioning via images and/or text. They encode the input image and/or textual prompt into a scene embedding, which is then used to condition a diffusion model that generates a Tri-Plane representing the scene.
- **LN3Diff** (Lan et al., 2024) enables fast text/single-image-to-3D generation. For single-image-to-3D, the image is encoded into a scene embedding, which is used to condition a DiT transformer that generates a Tri-Plane representing the scene.

Other works do change the rendering space of NeRFs, which makes them more closely related to our work, with still some key differences.

- **StyleNeRF** (Gu et al., 2022) proposes a high-resolution generative model with multi-view consistency. They achieve style-based generation via NeRFs that render feature images followed by a decoder. However, these features can only be obtained via the NeRF rendering procedure.
- **Latent-NeRF** (Metzer et al., 2023) tackles text-to-3D object generation. It trains latent NeRFs such that their renderings match the posterior distribution of Stable Diffusion (Rombach et al., 2022) under a descriptive text prompt. After this optimization, the latent NeRF is converted into an RGB NeRF to resolve the issues arising from training NeRFs in a standard latent space.
- **LERF** (Kerr et al., 2023) distills CLIP features in NeRFs while simultaneously learning the RGB components of the scene, which allows for language queries in 3D.

- **N3F** (Tschernezki et al., 2022) distills DINO features into a NeRF representation, while simultaneously learning the RGB components of the scene. This allows to apply image-level tasks (e.g. scene editing, object segmentation) on NeRF renderings. The DINO features rendered by NeRFs represent a 3D-consistent version of the original DINO features.

- **Latent-Editor** (Khalid et al., 2023) and **ED-NeRF** (Park et al., 2024) achieve scene editing by training NeRFs in a VAE latent space. To make this possible, they implement special layers ("adapter" and "refinement layer") that correct the NeRF renderings to match latent images.

- **RLS** (Aumentado-Armstrong et al., 2023) is the closest to our work. It employs hybrid NeRFs that are trained to simultaneously render both RGB and latent components of a scene, where the latent components are learned via a VAE decoder. At inference, it exclusively renders the latent components. However, due to the lack of a 3D-aware latent space, it still requires RGB components at training time to supervise the geometry of the scene.

The presented methods either tackle a *different task* or use a *different rendering space*, **which makes them not directly comparable with our work**. This means that our work is the first to propose training *purely* latent NeRFs, where no RGB components are utilized, for the reconstructive task.

## B  IMPLEMENTATION DETAILS

### B.1  NERFSTUDIO EXTENSION

We extend Nerfstudio to support latent scene learning. Specifically, we incorporate our latent NeRF training pipeline as well as IG-AE in Nerfstudio while adhering to its coding conventions, thereby enabling the training of any Nerfstudio model in our 3D-aware latent space. We modify the ray batching process to make it correspond to full images, where the default behavior uses randomly sampled pixels. This is necessary to be able to decode the rendered images without breaking the gradient flow. Beyond reproducibility purposes for this paper, we hope the proposed generic extension will facilitate research on latent NeRFs and their applications. Our code is open-source and available on the following GitHub repository: `https://github.com/AntoineSchnepf/latent-nerfstudio`.

### B.2  NERFSTUDIO MODELS TRAINING DETAILS

We test the following models from Nerfstudio: Vanilla-NeRF (Mildenhall et al., 2020), Instant-NGP (Müller et al., 2022), TensoRF (Chen et al., 2022), and K-Planes (Fridovich-Keil et al., 2023). For each method, the Nerfstudio framework provides a proprietary loss $F_\theta$, as well as a custom training procedure that includes specific optimizers, schedulers and other method-specific components. To train a latent NeRF in Nerfstudio, we first train the chosen model for $10\,000$ iterations to minimize $\mathcal{L}_{\text{LS}}$ using the method-specific optimization process. Subsequently, we continue the training with $15\,000$ iterations of RGB alignment by minimizing $\mathcal{L}_{\text{align}}$. To account for the change of image representations, we modulate the learning rate of each method by a factor of $\xi_{\text{LS}}$ in latent supervision, and a factor $\xi_{\text{align}}$ for RGB alignment. Appendix F.2 details the hyper-parameters we used in Nerfstudio, including the values of these factors for each method.

### B.3  IG-AE TRAINING DETAILS

We adopt the pre-trained "Ostris KL-f8-d16" VAE (Burkett, 2024) from Hugging Face, which has a downscale factor $l = 8$, and $c = 16$ feature channels in the latent space. We apply a Total Variation (TV) regularization on our Tri-Planes with a factor $\lambda_{\text{TV}}^{\text{3D}}$ which prevents overfitting on train views (we exclude it from Eq. (8) for clarity). Analogously, we also apply TV regularization on encoded real images $k_j$. A detailed presentation of TV and its benefits for NVS can be found in Appendix C. Training IG-AE takes 60 hours on $4\times$ NVIDIA L4 GPUs. Our detailed hyper-parameter settings can be found in Appendix F.1. The training code for IG-AE is open-source and available on the following GitHub repository: `https://github.com/k-kassab/igae`.

## C  Tri-Plane Representations

Tri-Plane representations (Chan et al., 2022) are explicit-implicit scene representations enabling scene modeling in three axis-aligned orthogonal feature planes, each of resolution $K \times K$ with feature dimension $F$. To query a 3D point $x \in \mathbb{R}^3$, it is projected onto each of the three planes to retrieve bilinearly interpolated feature vectors $F_{xy}$, $F_{xz}$ and $F_{yz}$. These feature vectors are then aggregated via summation and passed into a small neural network to retrieve the corresponding color and density, which are then used for volume rendering (Kajiya & Von Herzen, 1984). We adopt Tri-Plane representations for our set of latent scenes for their lightweight architectures and relatively fast training times.

**TV regularization.**    Total variation measures the spatial variation of images. It can act as a regularisation term to denoise images or ensure spatial smoothness. Total variation on images is expressed as follows:

$$\mathcal{L}_{TV}(I) = \frac{1}{HW} \sum_{i,j} [\|I_{i,j} - I_{i-1,j}\|_p^q + \|I_{i,j} - I_{i,j-1}\|_p^q] , \tag{13}$$

where $\|\cdot\|_p^q$ is the $l_p$ norm to the power $q$.

Total variation regularization was adapted for inverse graphics to promote spatial smoothness in implicit-explicit NeRF representation (Fridovich-Keil et al., 2023; 2022; Chen et al., 2022). Accordingly, we regularize our Tri-Planes with total variation, which we write below:

$$\mathcal{L}_{TV}^{3D}(T) = \sum_c \mathcal{L}_{TV}(T^{(c)}) , \tag{14}$$

where $T^{(c)}$ represents the feature plane of index $c$ in $T$. In practice, we add the regularization term $\lambda_{TV}^{3D} \sum_i \mathcal{L}_{TV}^{3D}(T_i)$ to our 3D regularization loss $\mathcal{L}_{3D}$ (Eq. (8)), where $\lambda_{TV}^{3D}$ is a hyperparameter. While this TV regularization is done on plane features, it effectively translates to the latent 3D scene, as latents are obtained from decoding these regularized features. This leads to smooth latent renderings and smooth gradients, as we utilize TV regularization here with $(p, q) = (2, 2)$, which discourages high frequency variations in features.

Additionally, we use total variation in our AE preservation loss $\mathcal{L}_{AE}$ (Eq. (10)) with $(p, q) = (2, 1)$ to discourage our autoencoder from producing latents with high frequency variations, while conserving sharp edges (Rudin et al., 1992). In fact, we noticed during our experiments that solely using an $l_2$ reconstructive objective on an autoencoder leads to latents with high frequencies. These high frequencies are inconsistent across latent encodings of the same 3D-consistent object. Hence, to ensure the compatibility of $\mathcal{L}_{ae}^{(real)}$ with the 3D regularization term $\mathcal{L}_{3D}$, we add a TV regularization on encoded latents.

## D  Supplementary Results

### D.1  Importance of RGB Alignment

We show in Table 4 the importance of the second stage of RGB Alignment in the Latent NeRF Training Pipeline, where NVS performances in the RGB space improves after aligning latent scenes with the RGB views.

### D.2  Additional Quantitative Results

Tables 5 and 10 to 14 illustrate additional quantitative evaluations of various NeRF architectures trained in a standard AE latent space and our IG-AE 3D-aware latent space on held-out Objaverse scenes, using our latent NeRF training pipeline. These results further support our conclusions. Additionally, Table 5 also includes our ablations.

Table 6 compares the NVS performance of latent NeRF methods on latent images. Specifically, it compares NeRF-rendered latent images with encoded latent images. Methods trained in the latent space of our IG-AE exhibit significantly better latent NVS performance, confirming its 3D-awareness.

Table 7 compares the autoencoding performance of our IG-AE and the baseline AE when used on images from held-out Objaverse scenes and OOD Shapenet scenes. Our IG-AE showcases better PSNR and SSIM metrics but slightly worse LPIPS metrics, which indicates a minor loss in high frequency details.

Table 8 compares RGB NeRFs and Latent NeRFs using IG-AE, on various Nerfstudio methods, in terms of NVS performance as well as training and rendering times.

### D.3 ADDITIONAL QUALITATIVE RESULTS

Figs. 6 to 8 compares the renderings of NeRFs respectively trained on the Cake, House and Figurine scene from Objaverse using our ablated models as well as our full model. Latent NeRFs trained with IG-AE (no 3D) present artifacts when decoding their renderings to the RGB space, as the latent space is not 3D-aware. IG-AE (no Pr) and IG-AE both have 3D-aware latent space and present no artifacts, as previously illustrated. IG-AE (no Pr) demonstrates degraded auto-encoding performances.

Fig. 9 illustrates real images that are auto-encoded using all IG-AE ablations.

Figs. 10 to 14 showcase additional qualitative results on held-out objaverse scenes.

Novel view synthesis consistency is better visualized in video format, which showcases frame-to-frame consistency. We provide such visualizations on our project page: https://ig-ae.github.io .

## E CHOICE OF AUTOENCODER

Table 9 compares autoencoders with various latent resolutions. For this comparison, we adopt the autoencoders "kl-f4", "kl-f8", "kl-f16" from (Rombach et al., 2022), exhibiting downscale factors of 4, 8, and 16 respectively. Note that the compared autoencoders are not trained into inverse graphics autoencoders to conduct this study. As the table presents, NVS performances are directly correlated with latent resolution. We speculate that this comes from the fact that a higher resolution latent space encourages a more local dependency between the RGB image and its latent representation, which is more advantageous in the context of 3D-awareness. As a baseline for IG-AE, we select an autoencoder with a downscale factor of 8 as a compromise between latent resolution and rendering speed.

## F HYPERPARAMETERS

This section details our hyperparameter settings for both our IG-AE training and our Latent NeRF training in Nerfstudio.

### F.1 IG-AE TRAINING SETTINGS

Table 15 details the hyperparameters taken to train our IG-AE.

### F.2 NERFSTUDIO TRAINING SETTINGS

Table 16 details the hyperparameters taken to train NeRF models in Nerfstudio.

Table 4: **RGB alignment value.** We show our NVS performances after latent supervision and RGB alignment when training NeRF models with IG-AE. NVS has better performances in the RGB space after doing our stage 2 of RGB alignment. All results are obtained on the Cake scene from the Objaverse dataset.

| | Latent Supervision | | | RGB Alignment | | |
|---|---|---|---|---|---|---|
| | PSNR | SSIM | LPIPS | PSNR | SSIM | LPIPS |
| Vanilla-NeRF | 24.34 | 0.870 | 0.175 | 34.68 | 0.967 | 0.050 |
| Instant-NGP | 22.08 | 0.826 | 0.182 | 28.41 | 0.917 | 0.063 |
| TensoRF | 24.12 | 0.862 | 0.175 | 33.06 | 0.962 | 0.043 |
| K-Planes | 22.68 | 0.842 | 0.200 | 33.20 | 0.962 | 0.053 |

Table 5: **Supplementary evaluations.** Quantitative NVS results on held-out scenes from Objaverse. This table extends Table 2 by conducting our ablations on all our adopted representations.

| | | Cake | | | House | | | Figurine | | |
|---|---|---|---|---|---|---|---|---|---|---|
| | | PSNR | SSIM | LPIPS | PSNR | SSIM | LPIPS | PSNR | SSIM | LPIPS |
| Vanilla-NeRF | AE | 33.18 | 0.962 | 0.053 | 31.71 | 0.946 | 0.031 | 31.46 | 0.972 | 0.017 |
| | IG-AE (no Pr) | 33.87 | 0.966 | 0.050 | 34,08 | 0.961 | 0.022 | 33.73 | 0.9821 | 0.012 |
| | IG-AE (no 3D) | 32.09 | 0.951 | 0.063 | 30,56 | 0,923 | 0,046 | 30.94 | 0.9650 | 0.021 |
| | IG-AE | 34.68 | 0.967 | 0.050 | 33,10 | 0,954 | 0,027 | 33.10 | 0.979 | 0.013 |
| Instant-NGP | AE | 24.69 | 0.860 | 0.122 | 22.67 | 0.784 | 0.196 | 24.62 | 0.908 | 0.086 |
| | IG-AE (no Pr) | 28.04 | 0.917 | 0.052 | 27,22 | 0,882 | 0,055 | 26.46 | 0.9297 | 0.043 |
| | IG-AE (no 3D) | 27.17 | 0.904 | 0.087 | 23,34 | 0,805 | 0,189 | 24.48 | 0.9078 | 0.091 |
| | IG-AE | 28.41 | 0.917 | 0.063 | 27,04 | 0,882 | 0,064 | 27.20 | 0.941 | 0.040 |
| TensoRF | AE | 28.79 | 0.928 | 0.046 | 27.31 | 0.904 | 0.066 | 27.60 | 0.949 | 0.037 |
| | IG-AE (no Pr) | 34.30 | 0.966 | 0.041 | 32,92 | 0,952 | 0,025 | 32.86 | 0.9780 | 0.012 |
| | IG-AE (no 3D) | 29.56 | 0.936 | 0.094 | 27,99 | 0,891 | 0,083 | 28.89 | 0.9515 | 0.034 |
| | IG-AE | 33.06 | 0.962 | 0.043 | 31,78 | 0,951 | 0,026 | 31.36 | 0.972 | 0.018 |
| K-Planes | AE | 31.82 | 0.954 | 0.056 | 30.06 | 0.927 | 0.038 | 30.45 | 0.965 | 0.021 |
| | IG-AE (no Pr) | 34.75 | 0.966 | 0.049 | 33,22 | 0,949 | 0,026 | 33.67 | 0.9796 | 0.011 |
| | IG-AE (no 3D) | 29.47 | 0.935 | 0.103 | 29,75 | 0,902 | 0,088 | 30.20 | 0.9602 | 0.027 |
| | IG-AE | 33.53 | 0.963 | 0.052 | 31,73 | 0,940 | 0,031 | 32.44 | 0.9751 | 0.014 |

Table 6: **Latent NVS performance.** Quantitative evaluation of NVS performance of NeRF methods on latent images. The metrics compare NeRF-rendered latent images with encoded latent images. Methods trained in the latent space of our IG-AE exhibit significantly better latent NVS performance, confirming its 3D-awareness.

| | | Objaverse (cake) | | Shapenet (vase) | |
|---|---|---|---|---|---|
| | | PSNR | SSIM | PSNR | SSIM |
| Vanilla-NeRF | AE | 38.36 | 0.854 | 38.48 | 0.841 |
| | IG-AE | **50.31** | **0.989** | **48.20** | **0.979** |
| Instant-NGP | AE | 35.87 | 0.768 | 35.94 | 0.734 |
| | IG-AE | **44.11** | **0.967** | **43.84** | **0.955** |
| TensoRF | AE | 36.79 | 0.817 | 36.56 | 0.780 |
| | IG-AE | **47.37** | **0.983** | **46.34** | **0.967** |
| K-Planes | AE | 36.85 | 0.798 | 36.53 | 0.755 |
| | IG-AE | **44.74** | **0.969** | **44.26** | **0.949** |

Table 7: **Synthetic reconstruction.** Comparison of autoencoding performances of IG-AE and the baseline autoencoder (AE) on images from Objaverse held-out scenes and Shapenet OOD scenes. In each dataset, the metrics are averaged over the images of all the test scenes.

|  | Objaverse | | | Shapenet | | |
|---|---|---|---|---|---|---|
|  | PSNR | SSIM | LPIPS | PSNR | SSIM | LPIPS |
| AE | 36.47 | 0.975 | **0.005** | 34.78 | 0.985 | **0.007** |
| IG-AE | **37.92** | **0.980** | 0.011 | **37.16** | **0.989** | 0.009 |

Table 8: **Comparison with RGB training.** Bringing NeRFs to the IG-AE latent space entails significantly lower training and rendering times for most methods. Rendering times represent an average over 1000 image renderings. RGB methods render images at a $128 \times 128$ resolution, whereas latent methods render at $16 \times 16$. Note that for latent NeRF methods, decoding takes an additional 6.7 ms for each method to obtain the final $128 \times 128$ images. Training and rendering time is measured using a single NVIDIA L4 GPU. As for rendering quality (NVS), some methods are more compatible with training in latent spaces than others, which we see as a direction of future improvements. NVS metrics are averaged over three Objaverse scenes: Cake, House and Figurine.

|  | Training Space | Training Time (min) | Rendering Time (ms) | NVS | | |
|---|---|---|---|---|---|---|
|  |  |  |  | PSNR | SSIM | LPIPS |
| Vanilla NeRF | RGB | 637 | 1201 | 40.78 | 0.993 | 0.003 |
|  | IG-AE | 28 | 19.7 | 33.60 | 0.966 | 0.030 |
| Instant NGP | RGB | 6 | 11.3 | 37.78 | 0.985 | 0.006 |
|  | IG-AE | 19 | 7.3 | 27.47 | 0.913 | 0.056 |
| TensoRF | RGB | 92 | 101.4 | 40.65 | 0.992 | 0.003 |
|  | IG-AE | 20 | 10.4 | 32.04 | 0.961 | 0.029 |
| K-Planes | RGB | 88 | 63.3 | 32.98 | 0.964 | 0.027 |
|  | IG-AE | 29 | 11.3 | 32.36 | 0.958 | 0.033 |

Table 9: **Autoencoder choice.** Comparison of the performance of autoencoders with various downscale factors on the task of latent NeRF training. We choose an autoencoder with an intermediate downscale factor of 8 as a compromise between latent NVS performance and training time.

|  | kl-f16 | | | kl-f8 | | | kl-f4 | | |
|---|---|---|---|---|---|---|---|---|---|
|  | PSNR | SSIM | LPIPS | PSNR | SSIM | LPIPS | PSNR | SSIM | LPIPS |
| Vanilla-NeRF | 29.58 | 0.941 | 0.047 | 32.22 | 0.962 | 0.027 | 35.38 | 0.978 | 0.013 |
| Instant-NGP | 23.54 | 0.844 | 0.117 | 26.16 | 0.900 | 0.084 | 30.03 | 0.948 | 0.036 |
| TensoRF | 25.91 | 0.891 | 0.080 | 28.46 | 0.934 | 0.042 | 33.56 | 0.972 | 0.014 |
| K-Planes | 27.59 | 0.917 | 0.060 | 30.06 | 0.943 | 0.032 | 33.42 | 0.966 | 0.017 |

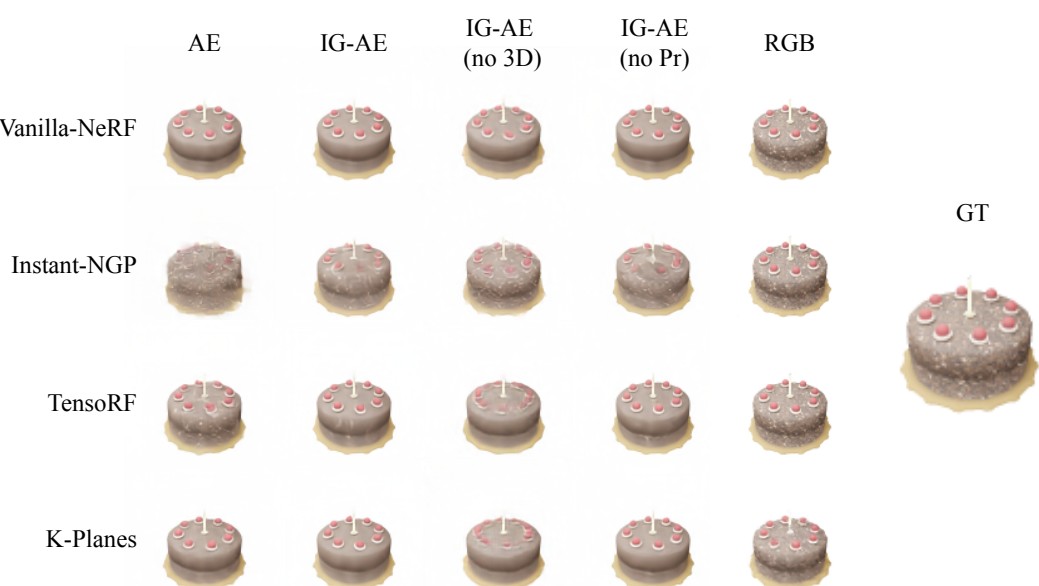

Figure 6: **Qualitative comparison.** NeRF renderings of the Cake scene from Objaverse.

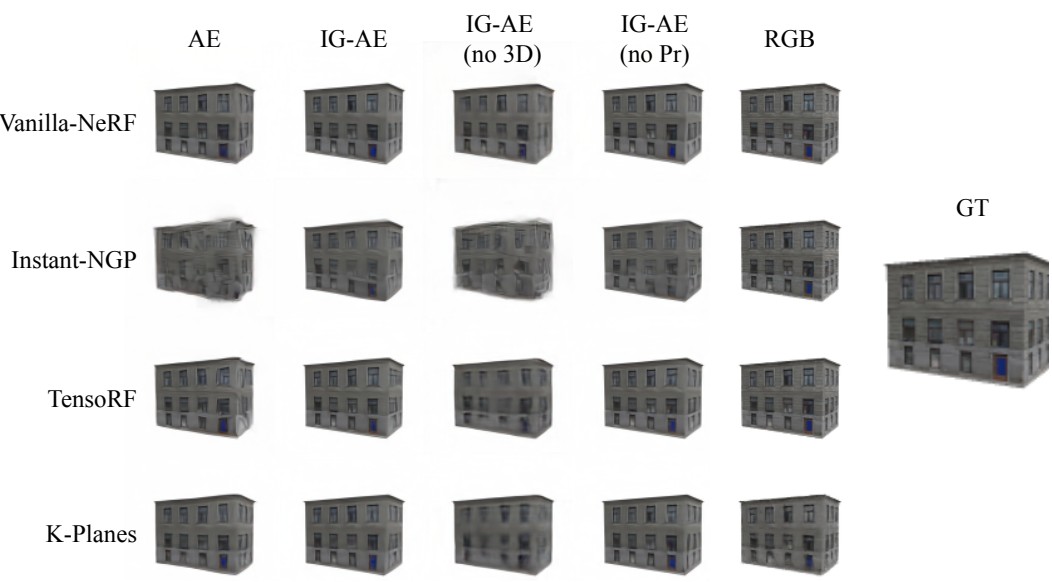

Figure 7: **Qualitative comparison.** NeRF renderings of the House scene from Objaverse.

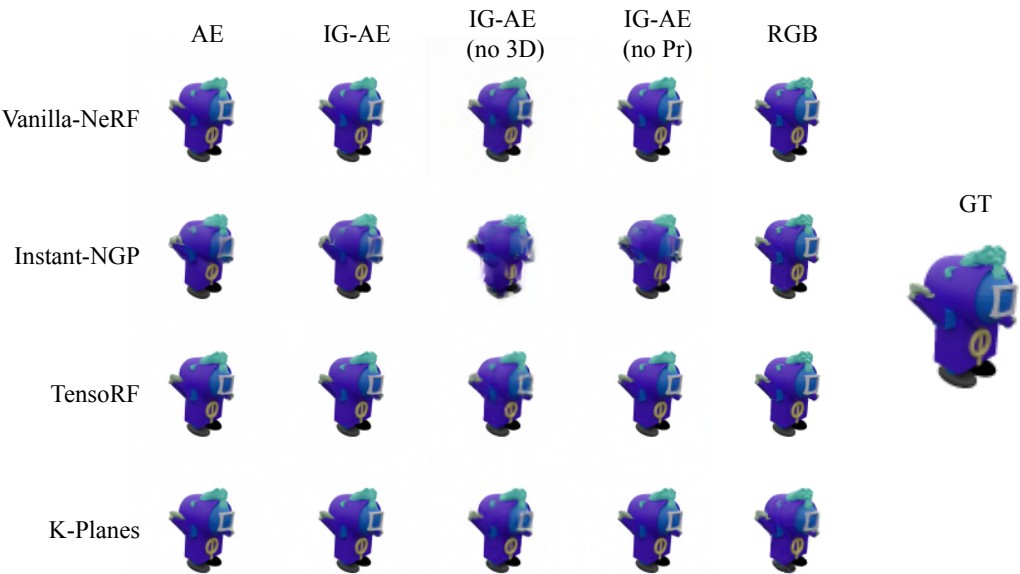

Figure 8: **Qualitative comparison.** NeRF renderings of the Figurine scene from Objaverse.

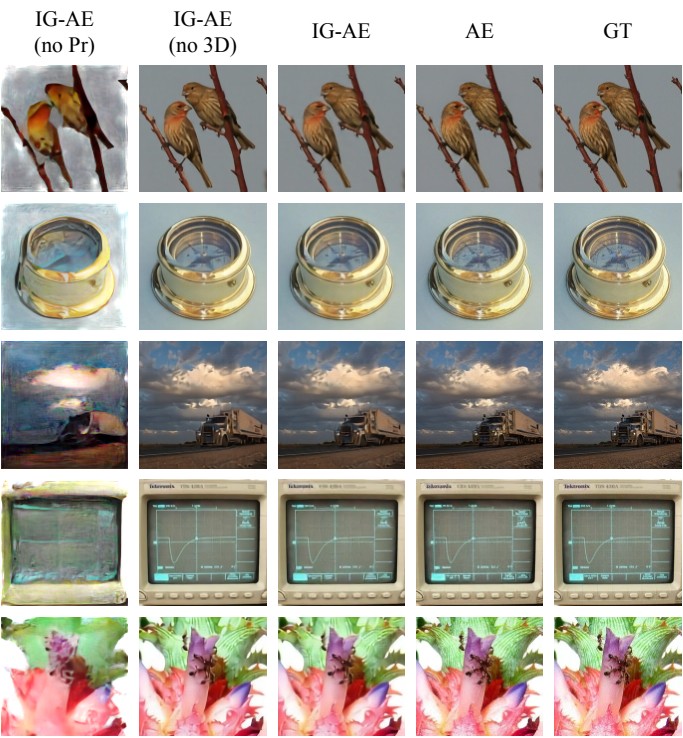

Figure 9: **Qualitative comparison of reconstruction quality when autoencoding.** IG-AE (no Pr) presents artifacts when auto-encoding images. IG-AE (no 3D) omits these artifacts but does not embed a 3D-aware latent space. IG-AE omits the artifacts present in IG-AE (no Pr), while still integrating a 3D-aware latent space.

Table 10: **Supplementary evaluations.** Quantitative NVS results on held-out scenes from Objaverse.

|  |  | Helmet | | | Book | | | Burger | | |
|---|---|---|---|---|---|---|---|---|---|---|
|  |  | PSNR | SSIM | LPIPS | PSNR | SSIM | LPIPS | PSNR | SSIM | LPIPS |
| Vanilla-NeRF | AE | 32.42 | 0.973 | 0.024 | 30.55 | 0.967 | 0.044 | 33.21 | 0.968 | **0.037** |
|  | IG-AE | **33.24** | **0.977** | **0.021** | **32.02** | **0.973** | **0.039** | **34.23** | **0.972** | **0.037** |
| Instant-NGP | AE | 24.63 | 0.918 | 0.087 | 23.61 | 0.904 | 0.144 | 23.52 | 0.862 | 0.123 |
|  | IG-AE | **28.27** | **0.951** | **0.035** | **27.22** | **0.950** | **0.058** | **27.48** | **0.922** | **0.044** |
| TensoRF | AE | 28.45 | 0.954 | 0.040 | 26.79 | 0.947 | 0.070 | 27.55 | 0.931 | 0.038 |
|  | IG-AE | **32.27** | **0.972** | **0.021** | **30.36** | **0.968** | **0.041** | **31.70** | **0.962** | **0.022** |
| K-Planes | AE | 31.14 | 0.966 | 0.027 | 28.90 | 0.959 | 0.051 | 31.91 | 0.958 | 0.040 |
|  | IG-AE | **33.27** | **0.975** | **0.019** | **31.42** | **0.970** | **0.039** | **33.53** | **0.968** | **0.037** |

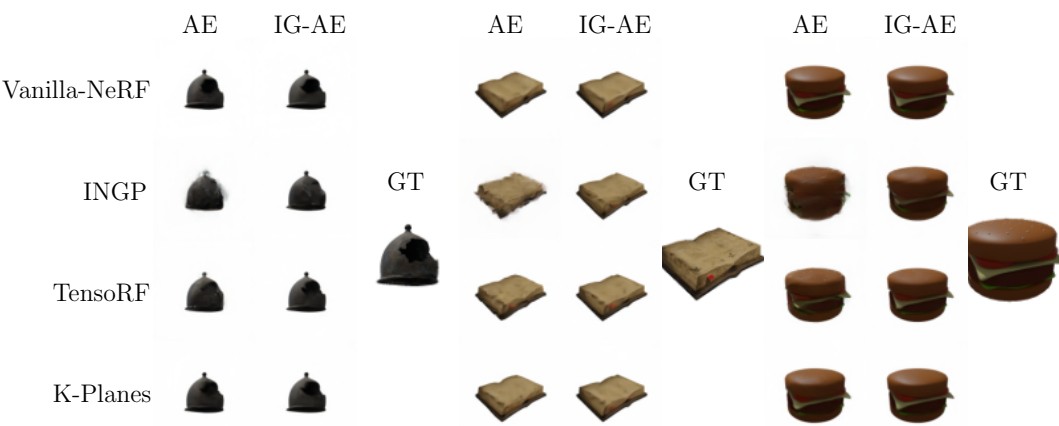

Figure 10: **Supplementary evaluations.** Qualitative NVS results on held-out scenes from Objaverse.

Table 11: **Supplementary evaluations.** Quantitative NVS results on held-out scenes from Objaverse.

|  |  | Cartoon | | | Dragon | | | Flower | | |
|---|---|---|---|---|---|---|---|---|---|---|
|  |  | PSNR | SSIM | LPIPS | PSNR | SSIM | LPIPS | PSNR | SSIM | LPIPS |
| Vanilla-NeRF | AE | 29.54 | 0.974 | 0.017 | 32.20 | 0.981 | 0.022 | 35.59 | 0.984 | 0.014 |
|  | IG-AE | **30.99** | **0.981** | **0.012** | **33.99** | **0.987** | **0.015** | **37.62** | **0.989** | **0.010** |
| Instant-NGP | AE | 22.58 | 0.906 | 0.104 | 25.32 | 0.932 | 0.108 | 27.66 | 0.925 | 0.091 |
|  | IG-AE | **25.68** | **0.949** | **0.038** | **28.42** | **0.961** | **0.044** | **31.07** | **0.963** | **0.035** |
| TensoRF | AE | 25.44 | 0.946 | 0.045 | 28.53 | 0.964 | 0.044 | 30.62 | 0.962 | 0.038 |
|  | IG-AE | **29.29** | **0.973** | **0.017** | **32.65** | **0.983** | **0.017** | **35.44** | **0.985** | **0.013** |
| K-Planes | AE | 28.22 | 0.967 | 0.021 | 31.09 | 0.976 | 0.027 | 33.77 | 0.978 | 0.023 |
|  | IG-AE | **30.57** | **0.979** | **0.012** | **33.24** | **0.984** | **0.016** | **37.26** | **0.988** | **0.011** |

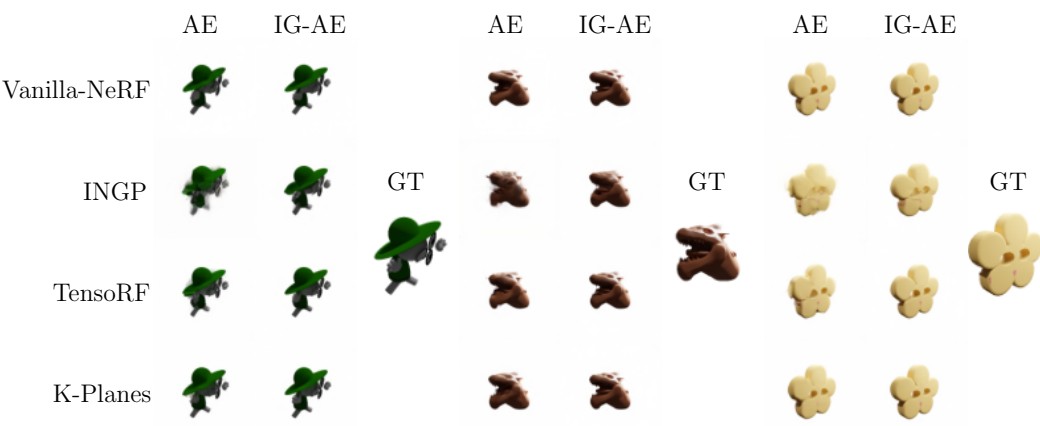

Figure 11: **Supplementary evaluations.** Qualitative NVS results on held-out scenes from Objaverse.

Table 12: **Supplementary evaluations.** Quantitative NVS results on held-out scenes from Objaverse.

|  |  | Gravestone | | | Hut | | | Painting | | |
|---|---|---|---|---|---|---|---|---|---|---|
|  |  | PSNR | SSIM | LPIPS | PSNR | SSIM | LPIPS | PSNR | SSIM | LPIPS |
| Vanilla-NeRF | AE | 31.64 | 0.973 | 0.034 | 30.33 | 0.963 | 0.030 | 36.07 | 0.984 | 0.018 |
|  | IG-AE | **32.63** | **0.978** | **0.029** | **30.70** | **0.964** | **0.027** | **37.79** | **0.989** | **0.013** |
| Instant-NGP | AE | 22.95 | 0.885 | 0.169 | 22.64 | 0.881 | 0.154 | 27.02 | 0.930 | 0.089 |
|  | IG-AE | **26.68** | **0.941** | **0.073** | **26.04** | **0.933** | **0.066** | **34.34** | **0.980** | **0.020** |
| TensoRF | AE | 26.65 | 0.944 | 0.076 | 26.89 | 0.943 | 0.058 | 31.51 | 0.967 | 0.037 |
|  | IG-AE | **31.15** | **0.974** | **0.032** | **30.00** | **0.963** | **0.030** | **37.75** | **0.989** | **0.012** |
| K-Planes | AE | 28.72 | 0.960 | 0.053 | 28.68 | 0.955 | 0.038 | 34.79 | 0.982 | 0.020 |
|  | IG-AE | **31.25** | **0.973** | **0.031** | **30.19** | **0.961** | **0.029** | **38.28** | **0.990** | **0.012** |

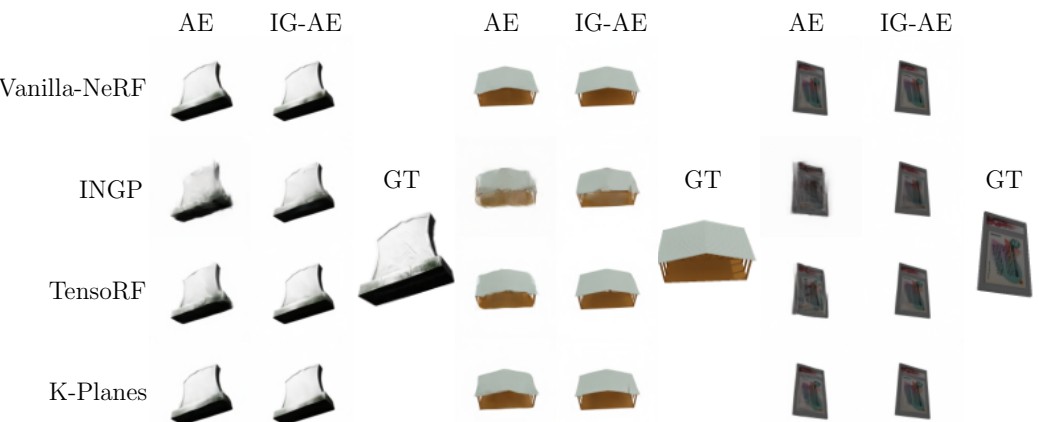

Figure 12: **Supplementary evaluations.** Qualitative NVS results on held-out scenes from Objaverse.

Table 13: **Supplementary evaluations.** Quantitative NVS results on held-out scenes from Objaverse.

| | | Pig | | | Ringer | | | Robot | | |
|---|---|---|---|---|---|---|---|---|---|---|
| | | PSNR | SSIM | LPIPS | PSNR | SSIM | LPIPS | PSNR | SSIM | LPIPS |
| Vanilla-NeRF | AE | 33.21 | 0.981 | 0.029 | 32.14 | 0.975 | 0.024 | 29.94 | 0.962 | 0.028 |
| | IG-AE | **34.34** | **0.987** | **0.019** | **33.70** | **0.982** | **0.015** | **31.76** | **0.973** | **0.019** |
| Instant-NGP | AE | 24.61 | 0.913 | 0.140 | 24.68 | 0.901 | 0.119 | 23.49 | 0.884 | 0.107 |
| | IG-AE | **28.35** | **0.956** | **0.067** | **28.24** | **0.951** | **0.042** | **26.01** | **0.923** | **0.049** |
| TensoRF | AE | 28.61 | 0.959 | 0.060 | 27.71 | 0.947 | 0.057 | 26.43 | 0.933 | 0.042 |
| | IG-AE | **32.58** | **0.981** | **0.027** | **32.03** | **0.976** | **0.019** | **29.96** | **0.963** | **0.024** |
| K-Planes | AE | 31.03 | 0.972 | 0.044 | 30.61 | 0.964 | 0.032 | 29.08 | 0.954 | 0.031 |
| | IG-AE | **33.12** | **0.981** | **0.026** | **32.99** | **0.975** | **0.020** | **31.10** | **0.967** | **0.022** |

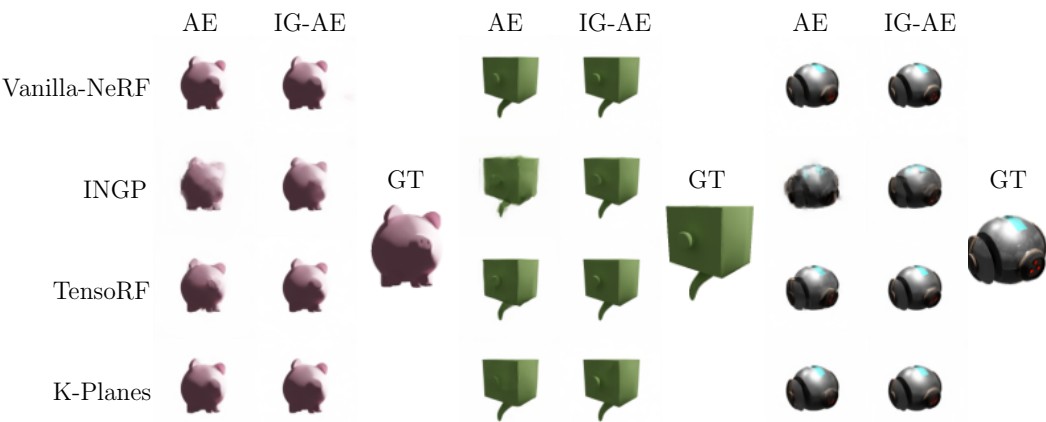

Figure 13: **Supplementary evaluations.** Qualitative NVS results on held-out scenes from Objaverse.

Table 14: **Supplementary evaluations.** Quantitative NVS results on held-out scenes from Objaverse.

| | | Coffin | | | Table | | | Villa | | |
|---|---|---|---|---|---|---|---|---|---|---|
| | | PSNR | SSIM | LPIPS | PSNR | SSIM | LPIPS | PSNR | SSIM | LPIPS |
| Vanilla-NeRF | AE | 30.61 | 0.968 | 0.042 | 33.06 | 0.978 | 0.024 | 30.72 | 0.970 | 0.033 |
| | IG-AE | **33.17** | **0.976** | **0.032** | **34.96** | **0.983** | **0.019** | **32.18** | **0.977** | **0.024** |
| Instant-NGP | AE | 21.67 | 0.876 | 0.211 | 23.29 | 0.873 | 0.161 | 24.21 | 0.903 | 0.138 |
| | IG-AE | **27.25** | **0.941** | **0.071** | **29.19** | **0.956** | **0.042** | **26.18** | **0.935** | **0.084** |
| TensoRF | AE | 27.35 | 0.946 | 0.076 | 27.16 | 0.943 | 0.071 | 26.86 | 0.943 | 0.065 |
| | IG-AE | **31.87** | **0.973** | **0.034** | **33.20** | **0.979** | **0.020** | **30.59** | **0.971** | **0.031** |
| K-Planes | AE | 29.30 | 0.960 | 0.053 | 30.72 | 0.965 | 0.036 | 28.86 | 0.957 | 0.048 |
| | IG-AE | **31.56** | **0.971** | **0.036** | **33.25** | **0.976** | **0.023** | **31.66** | **0.972** | **0.027** |

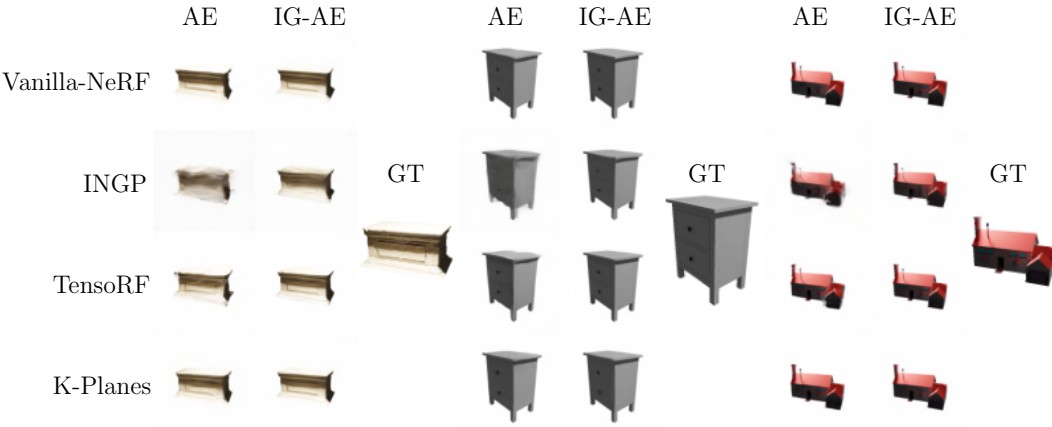

Figure 14: **Supplementary evaluations.** Qualitative NVS results on held-out scenes from Objaverse.

Table 15: **IG-AE hyperparameters.** Hyperparameters used for our IG-AE training. More detailed information can be found in the configuration files of our open-source code.

| Parameter | Value |
|---|---|
| General | |
| Number of scenes $N$ | 500 |
| Pretraining epochs | 50 |
| Training epochs | 75 |
| Loss | |
| $\lambda_{\text{latent}}$ | 1 |
| $\lambda_{\text{RGB}}$ | 1 |
| $\lambda_{\text{TV}}^{\text{3D}}$ | $1 \times 10^{-4}$ |
| $\lambda_{\text{ae}}^{(\text{synth})}$ | 0.1 |
| $\lambda_{\text{ae}}^{(\text{real})}$ | 0.1 |
| $\lambda_{\text{p}}$ | 0.1 |
| $\lambda_{\text{TV}}$ | $1 \times 10^{-4}$ |
| Optimization | |
| Optimizer | Adam |
| Batch size (scene views) | 12 |
| Batch size (real images) | 3 |
| Learning rate (encoder) | $5 \times 10^{-5}$ |
| Learning rate (decoder) | $5 \times 10^{-5}$ |
| Learning rate (Tri-Planes) | $1 \times 10^{-4}$ |
| Scheduler | Exponential Decay |
| Decay factor | 0.988 |

Table 16: **Latent NeRF training hyperparameters.** Hyperparameters taken to train our Latent NeRFs in Nerfstudio. More detailed information can be found in the configuration files of our open-source Nerfstudio extension.

| NeRF model | Parameter | Value |
|---|---|---|
| Any | Latent Supervision iterations | 10 000 |
| | RGB Alignment iterations | 15 000 |
| | Batch size | 4 |
| | NeRF learning rate | Nerfstudio default |
| | NeRF optimizer | Nerfstudio default |
| | NeRF scheduler | Nerfstudio default |
| | Decoder learning rate | $1 \times 10^{-4}$ |
| | Decoder optimizer | Adam |
| | Decoder scheduler | Exponential decay |
| | Decoder decay factor | 0.9996 |
| Vanilla NeRF | $\xi_{\mathrm{LS}}$ | 0.1 |
| | $\xi_{\mathrm{align}}$ | 1 |
| Instant-NGP | $\xi_{\mathrm{LS}}$ | 0.001 |
| | $\xi_{\mathrm{align}}$ | 0.1 |
| TensoRF | $\xi_{\mathrm{LS}}$ | 0.01 |
| | $\xi_{\mathrm{align}}$ | 0.1 |
| K-Planes | $\xi_{\mathrm{LS}}$ | 0.1 |
| | $\xi_{\mathrm{align}}$ | 0.1 |

