# OpenReview forum: "Bringing NeRFs to the Latent Space: Inverse Graphics Autoencoder"
_ICLR.cc/2025/Conference — ICLR 2025 Poster_

### Official Review · Reviewer_iRnM · 2024-10-27

**Soundness:** 2
**Presentation:** 2
**Contribution:** 2
**Rating:** 6
**Confidence:** 4

**Summary:**

This paper introduces an Inverse Graphics Autoencoder (IG-AE) to explore the potential of inverse graphics in 2D latent spaces, which is relatively underutilized in computer vision. The authors highlight that inverse graphics applied in latent spaces can reduce both training and rendering complexity while enabling compatibility with other 2D latent-based methods. A key challenge addressed is that traditional image latent spaces lack 3D geometry, preventing direct application of inverse graphics. IG-AE addresses this by aligning the latent space of an image autoencoder with jointly trained latent 3D scenes. This enables a "latent NeRF" training pipeline, implemented within the Nerfstudio framework, making latent scene learning accessible for methods supported by the framework. Experiments show that latent NeRFs trained with IG-AE offer improved quality compared to conventional autoencoders, while also providing faster training and rendering compared to NeRFs trained in image space.

**Strengths:**

1. **Novel Use of Inverse Graphics in Latent Space**: The authors claim that they are the first to explore this direction, which explores the relatively untapped area of applying inverse graphics in 2D latent spaces, which reduces training and rendering complexity and offers compatibility with other latent-based 2D methods.

2. **Integration of 3D Geometry into Latent Spaces**: The authors address the issue of the lack of 3D geometry in standard image latent spaces by regularizing the autoencoder with 3D information, aligning its latent space with jointly trained 3D scenes. This approach enhances the representation capability of latent spaces for 3D scene understanding. From many perspectives, these efforts will be meaningful in this area.

3. **Latent NeRF Training Pipeline**: Two new training pipelines are proposed. By creating a latent NeRF training pipeline and integrating it within the Nerfstudio framework, the paper unlocks efficient latent scene learning and makes it accessible to a broader set of methods, providing a practical tool for further research.

4. **Open-Source Contribution**: The open-source extension to the Nerfstudio framework adds value to the research community by facilitating reproducibility and further experimentation, and the proposed method can naturally work on any new NeRF representations.

**Weaknesses:**

1. **The writing is pretty hard to follow.** I tried to understand this paper by reading over and over again, but still find it very hard to follow. Since this direction is relatively new, I strongly suggest the author to revise the writing in the later version for clearer elaboration.

2. **The motivation of the proposed method is unclear.** From my understanding, the proposed method try to align the NeRF rendering to a pretrained autoencoder (Ostris KL-f8-d16 VAE here), but what are the benefits of doing this? Though promising PSNR/LPIPS metrics are achieved as in Tab. 1 and Tab. 2, does the proposed IG-AE / AE supports zero-shot novel view synthesis on the imagenet dataset? Regarding Tab. 2, I don't quite understand how the author evaluate the novel view synthesis pipeline here, since no visual results are included in the main paper. All the visual results are just reconstructing the input view.

3. **Lack of comparison with existing methods**. Though the author claims this idea is quite novel and they are among the first to propose this solution, many existing methods have already shown similar spirits, including

* SRT: Scene Representation Transformer.
* NeRF-VAE: A Geometry Aware 3D Scene Generative Model
* LN3Diff: Scalable Latent Neural Fields Diffusion for Speedy 3D Generation

Their 3D (V)AE models support 3D view synthesis on the open vocabulary single-view inputs. The discussions / comparisons with these methods are needed to demonstrate the soundness of the proposed pipeline.

**Questions:**

1. In section 3, is the same VAE used as in section 4? The implementation details have not mentioned this.
2. How is the "3D aligned" auto-encoder useful in the downstream tasks?
3. In Tab. 2, how is the NVS task performed? Given a view as the input to your encoder, then synthesis novel views directly? Would you include more visual results?
4. In your AE/VAE design, the latent space is still a "3D-aware" image, so why not using a truly 3D-aware latent such as the latent triplane in LN3Diff / Direct3D?

---

> ### Author Response · Authors · 2024-11-19
>
> We thank the reviewer for their constructive remarks. We address the reviewer's concerns about the motivation of our paper in our [**global response**](https://openreview.net/forum?id=LTDtjrv02Y&noteId=mkPwBGuY4r). In this response, we provide specific answers to the other concerns and questions of the reviewer.
>
> ### (w.1) Clarification & writing
>
> We understand that this new direction of research can present challenges for readers and are open to revising any part of the paper to improve is clarity. Nevertheless, as the other reviewers found our method clear (jLB8 s.2, RTXt s.4), it is difficult for us to propose any change without more details from the reviewer. Could the reviewer provide a more circumstantial commentary on the parts of the paper they found to be unclear?
>
> In any case, we provide some clarifications below, as the reviewer may have misinterpreted some messages in this work, particularly the evaluation protocol of our method.
>
> - This work employs NeRF to do Novel View Synthesis. Traditionally, this has been done by training them on many posed training RGB images, and evaluating them by comparing their renderings from held-out camera poses with held-out RGB views.
> - Our goal is to do the same operation but in the latent space of an autoencoder instead of the RGB space, using the decoder to then render held-out views from the latent to the RGB space.
> - To do so, we propose a "3D-aware latent space", which is a 2D (autoencoder) latent image space that respects the underlying geometry in images. Traditionally, latent spaces do not have this property, making them not directly compatible with NeRFs.
> - To enable latent NeRFs, we propose two main components: (i) a Latent NeRF Training Pipeline that enables training NeRFs in a latent space, and (ii) an IG-AE that encompasses a 3D-aware latent space that is more compatible with NeRFs.
> - Latent NeRFs are supervised by many views. Throughout the entire paper, the Novel View Synthesis performance is evaluated by rendering views from held-out camera poses, and comparing them with held-out ground truth images. The visual results in the paper represent views rendered from the held-out camera poses. They have never been seen during training.

---

> ### Author Response · Authors · 2024-11-19
>
> ### (w.2) Motivation
>
> We thank the reviewer for this remark, which helped us identify areas of improvement in the presentation of the motivation of the paper. We refer the reviewer to our [global response](https://openreview.net/forum?id=LTDtjrv02Y&noteId=mkPwBGuY4r), section "Paper Motivation".
>
> The reviewer then states that they do not understand how we evaluate novel view synthesis and that there is no visual results in the paper. They also claim that the visual results are just reconstructing the input view. This is incorrect. We clarify our evaluation setting as follows.
>
> **Our visual results in the paper illustrate held-out views** and have never been seen during training (they do not reconstruct input views). As clarified previously, **Novel View Synthesis is evaluated as traditionally done in the literature**, by comparing renderings from held-out camera poses with held-out views. Our NVS metrics are computed on these never-seen views in the RGB space. In any case, as the reviewer finds our results in Tables 1 and 2 to be "promising", we hope that they re-consider their evaluation in light of our clarifications.
>
> ### (w.3) Comparison with existing methods
>
> We thank the reviewer for their suggestions. We will take this into consideration when revising the related work section of our paper and explicitly highlight both the similarities and key distinctions.
>
> The suggested papers share similarities with our work, as they all use latent representations of the input image(s) as a part of a 3D model. However, key differences set them apart from our approach, making them unsuitable as baselines for comparison with our work:
>
> - **Different task.** The suggested papers address the task of few-view to 3D (SRT, L3Diff, NeRF-VAE) or 3D generation (NeRF-VAE). We address a more general problem: how can one design a NeRF-compatible autoencoder latent space? For this purpose, the specific task we are evaluating is novel view synthesis from numerous views by training NeRFs in the adapted latent space.
> - **Nature of Latent Representation.** Our latent representations of the training views are purely 2D latent images, which can be treated as regular images. This contrasts with the suggested papers, where the latent representations of the training images undergo a deeper structural transformation and are no longer in image form.
> - **Specificity to the NeRF representation**. In the suggested works, they design latent representations specifically for their adopted 3D representation. In contrast, our approach focuses on latent representations that are not specific to a particular NeRF representation.
>
> Finally, we would like to emphasize about a recurring confusion between our work and NeRF-VAE.
> - **NeRF-VAE.** NeRF-VAE can be misinterpreted as closely related to our work because it combines NeRFs and VAEs, but it is fundamentally different. Their model is a VAE that learns a distribution over radiance fields (that render in RGB space) by conditioning them on a latent representation $z$ of all the input views. In our case, we use an Image AE to modify the space in which classical NeRFs are trained (from RGB to latent space).
>
> ### Questions
>
> > 1. In section 3, is the same VAE used as in section 4? The implementation details have not mentioned this.
>
> Section 3 tackles latent NeRF learning (independently from the choice of AE). Section 4 proposes a new IG-AE that works better for learning latent NeRFs. Throughout the entire paper, "AE" refers to a standard autencoder, while "IG-AE" refers to our inverse graphics autoencoder.
>
> > 2. How is the "3D aligned" auto-encoder useful in the downstream tasks?
>
> An inverse graphics autoencoder would enable the application of  latent-based 2D method on 3D tasks by applying them on 3D-aware, but still 2D, latent images. We will make this clearer when revising the motivation of our work in the introduction.
>
> > 3. In Tab. 2, how is the NVS task performed? Given a view as the input to your encoder, then synthesis novel views directly? Would you include more visual results?
>
> Novel view synthesis is performed by training a latent NeRF using our Latent NeRF Training Pipeline on a set of posed images and then rendering novel latent images from held-out camera poses, which are then decoded to the RGB space. The visual results in our paper represent held-out views that have never been seen during training. We provide additional visual results here at the following link (https://anon-supp-iclr25.github.io/), where we display animated videos of our decoded latent renderings.
>
> > 4. In your AE/VAE design, the latent space is still a "3D-aware" image, so why not using a truly 3D-aware latent such as the latent triplane in LN3Diff / Direct3D?
>
> Our work specifically targets image autoencoders. Therefore, our latent representation are images.

---

> > ### Comment · Reviewer_iRnM · 2024-11-25
> > **Official comment**
> >
> > Dear authors,
> >
> > Thanks for clarifying most of my concerns, and I still have a few questions left:
> >
> > 1. When you are evaluating the novel view synthesis task, how does the proposed method render the novel view, is the IG-AE still used? Also, given a new object, what is the inference pipeline for the novel view synthesis. Train a triplane/k-plane/TensorF with the IG-AE fixed with the L_latent and L_rgb?
> > 2. I wonder whether the pre-trained VAE can be used in other down-stream tasks, e.g., since the encoded features are geometry aware, whether they can perform better on some 3D tasks such as establishing 3D dense correspondences from multi-view images.
> >
> > I lean to accept this paper and please revise the writing in the camera ready version to make it more sound and clearer.

---

> ### Author Response · Authors · 2024-11-25
>
> We would like to thank the reviewer for their response.
>
> > When you are evaluating the novel view synthesis task, how does the proposed method render the novel view, is the IG-AE still used?
>
> During inference, after a latent NeRF has been trained, we can render novel views in the latent space. We then use the decoder of IG-AE to decode the new rendered latent view back to the RGB space, providing the novel view in RGB.
>
> > Also, given a new object, what is the inference pipeline for the novel view synthesis. Train a triplane/k-plane/TensorF with the IG-AE fixed with the L_latent and L_rgb?
>
> Given a new object, we first need to train the latent NeRF. To do so, we apply the Latent NeRF Training Pipeline described in Section 3.2 (Latent NeRF Training) with IG-AE as the autoencoder. Figure 3 illustrate this training process, in which the "NeRF model" could be K-Planes or TensorRF, for instance. More technically, the training is done using L_LS and L_align (cf. section 3.2).
>
> For inference, one can perform novel view synthesis by sequentially:
> - rendering the latent NeRF,
> - decoding this rendering to the RGB space using the decoder part of IG-AE.
>
> This is illustrated in the "Latent NeRF Inference Pipeline" in Figure 3. Due to the 3D-aware nature of IG-AE, this decoding step preserves 3D consistency and image fidelity in the RGB space.
>
> > I wonder whether the pre-trained VAE can be used in other down-stream tasks, e.g., since the encoded features are geometry aware, whether they can perform better on some 3D tasks such as establishing 3D dense correspondences from multi-view images.
>
> In our paper, we only experimentally showcase the applicability of IG-AE for latent scene learning. Nonetheless, we believe that it may be used in other applications that require an image autoencoder, where the 3D-aware latents would help to improve the performance. This is a primary motivation of our work, which we better illustrate in our revised introduction. For instance, IG-AE may be better suited than the baseline VAE used in the following papers, leading to potential performance increase:
> - Latent-NeRF [1] perform scene generation by training latent NeRFs. Better quality could be achieved by training the latent NeRFs in the IG-AE latent space, thanks to its 3D-aware nature.
> - ED-NERF [2] and Latent-Editor [3] perform scene editing. To do so, they use latent NeRF with some workarounds (adapters and refinement layers) to ensure their compatibility with the used VAE. With IG-AE, they would not need such workarounds, which would simplify their pipeline and potentially improve their performances.
> - Marigold [4] for depth estimation (2D task that is 3D-aware and uses a 2D VAE).
>
> Also, the fact that our Latent NeRFs are trained in IG-AE latent space makes them compatible with 2D models like in [5] for scene segmentation.
>
> We are not aware of any methods utilizing image autoencoders to achieve *3D dense correspondences from multi-view images*. If such methods exist, we believe that IG-AE could provide some level of improvement.
>
> #### References
>
> [1] Gal Metzer, Elad Richardson, Or Patashnik, Raja Giryes, and Daniel Cohen-Or. Latent-NeRF for Shape-Guided Generation of 3D Shapes and Textures. In Proceedings of the IEEE/CVF Conference on Computer Vision and Pattern Recognition (CVPR), pp. 12663–12673, June 2023.
>
> [2] Park, J., Kwon, G., & Ye, J. C. (2024). ED-NeRF: Efficient Text-Guided Editing of 3D Scene With Latent Space NeRF. The Twelfth International Conference on Learning Representations.
>
> [3] Khalid, U., Iqbal, H., Karim, N., Hua, J., & Chen, C. (2023). LatentEditor: Text Driven Local Editing of 3D Scenes. arXiv Preprint arXiv:2312. 09313.
>
> [4] Ke, B., Obukhov, A., Huang, S., Metzger, N., Daudt, R. C., & Schindler, K. (2024, June). Repurposing Diffusion-Based Image Generators for Monocular Depth Estimation. Proceedings of the IEEE/CVF Conference on Computer Vision and Pattern Recognition (CVPR), 9492–9502.
>
> [5] Tian, J., Aggarwal, L., Colaco, A., Kira, Z., & Gonzalez-Franco, M. (2024, June). Diffuse Attend and Segment: Unsupervised Zero-Shot Segmentation using Stable Diffusion. Proceedings of the IEEE/CVF Conference on Computer Vision and Pattern Recognition (CVPR), 3554–3563.

---

> > ### Comment · Reviewer_iRnM · 2024-11-26
> > **Official reply to the author**
> >
> > Thanks for the clarification, now I gets a clearer picture of the proposed method. The revision pdf looks nice and improves the soundness of the proposed method.
> >
> > Regarding related works that bridge the 3D correspondences / features with multi-view/video inputs, the following works are also related and may also be discussed / compared in the final camera ready version:
> >
> > * Improving 2D Feature Representations by
> > 3D-Aware Fine-Tuning [ECCV 24]: https://ywyue.github.io/FiT3D/
> > * Probing the 3D Awareness of Visual Foundation Models [CVPR 24]: https://arxiv.org/abs/2404.08636
> >
> > I will keep my current rating (6) and lean to accept this paper.

---

> > > ### Author Response · Authors · 2024-11-26
> > >
> > > We thank the reviewer for their response.
> > >
> > > The suggested papers are indeed very interesting, and will be included in the camera-ready version of the paper.
> > >
> > > For now, this message is to inform the reviewer that we have submitted a **second revision** of our paper, in which we aim for **better clarity** on the choice of dataset, as well as the limitations of our method which is **explicitly discussed in a dedicated section** (changes in violet).
> > >
> > > We hope that these changes further improve our paper and better address the reviewer's concerns.
> > >
> > > We remain open for further discussions and suggestions.

---

> ### Author Response · Authors · 2024-11-25
> **Paper Revision**
>
> As promised, we have submitted our paper revision with the changes discussed in the rebuttal, highlighted by the blue parts in our revision. We have provided a global message [here](https://openreview.net/forum?id=LTDtjrv02Y&noteId=Yjcz1Gp7aq) to highlight the changes and added experiments in our paper.
>
> Particularly, to address the reviewer's remarks:
>
> - **(w.2)** We have revised the introduction to **improve writing** and to **better illustrate the motivation** of our work, as well as the advantages and applications latent NeRFs bring.
> - **(w.3)** We have **added a new section to our related work** that contrasts our work from other works tackling feature fields or scene embeddings in the literature (including NeRF-VAE and LN3Diff). Additionally, we have added a section and a summarizing table in the appendix, providing a more detailed illustration of recent works and highlighting their distinctions.
> - Overall, we have enhanced the writing in some sections and will refine it further in the camera-ready version of the paper.
>
> We remain open for further discussions.

---

### Official Review · Reviewer_jLB8 · 2024-10-30

**Soundness:** 3
**Presentation:** 2
**Contribution:** 2
**Rating:** 6
**Confidence:** 5

**Summary:**

This paper propose a two-stage pipeline to train a latent space which is suitable for fitting NeRFs. In the first stage, a set of latent tri-planes are individually trained for each scene in the training data. The latent tri-planes are fitted by minimizing both the volumetric rendering error in latent space and the pixel difference in RGB space through a image decoder. In the second stage, an autoencoder is trained on all scenes in the training data, jointly with all individual tri-planes. The fidelity of the autoencoder is additionally regularized by training on additional real images. Experiments demonstrated that the proposed latent space is able to fitting NeRFs better than a image-based latent space.

**Strengths:**

- The paper attempts to research on an important problem of 3D-aware latent spaces.

- The proposed method overall makes sense: the introduce of individually trained latent tri-planes provides a auxiliary variable which serves as a "3D-aware" guidance for auto-encoded latents.

- Experiments show that the proposed IG-AE is better for training NeRFs than vanilla AE.

- The method can easily integrated into NeRFStudio with an open-source extension.

**Weaknesses:**

My majority concerns are as follows:

(1) While the proposed method is interesting and makes sense, the claimed property of "3D-aware" latent space cannot be fully justified from the given experiment results:
- (a) The proposed method is only tested on dataset with limited variations. For evaluation NeRFs, ShapeNet dataset is not a best choice as it contains simple shapes and textures without any non-Lambertian effects. Additionally, the test dataset contains only three categories for Shapenet and Objeverse dataset. Evaulations on more complicated dataset with non-lambertian effects, such as Realistic Synthetic 360 [1] would help to more thoroughly demonstrate the "3D-aware" property of the latent space across a wider range of scenarios and make the results more convincing.
- (b) Even under the limited test dataset, the performance of the proposed method are not convincing enough. Despite the IG-AE outperforms the vanilla AE counterpart, the quality degrades significantly compares to the RGB version. This raises the question that whether the image fidelity is enough for the IG-AE.
- (c) The "3D-aware" property cannot be easily judged from given metrics and results in the paper. All quantitative results are averaged across views - it is hard to know whether the "consistency" between different synthesized views are preserved. One suggestion would be using some perceptal metrics to evaluate the consistency between different generated view, e.g., the CLIP feature similarity used in [2] and [3]. There are also very few qualitative results provided. Providing more qualitative results such as videos showing a rotated object can be also useful to visualize the view consistency.
- (d) The effect of the original auto-encoder utilized for training IG-AE is not well explored. As discussed in Section 3.2 in [4], a latent space with low channel-wise depth (e.g., down to 4) and a slightly higher (but still significantly lower than RGB input resolution, 32 or 64 for example) will encourage local dependency over the autoencoder’s image and latent spaces. Under such circumstance, the latent representation is a near patch level representation of its corresponding RGB image, making it nearly equivariant to spatial transformations of the scene. In other words, it automatically (at least to some extent) has the property of "3D-aware" for training NeRFs. I would suggest the paper add more discussion and/or experiments regarding the 2D autoencoder used.

(2) There has been many attempts for training NeRFs (or similar implicit 3D representations) in a space with reduced resolution, followed with upsampler operations in 2D space. For example:
- Rodin [5] first generates a low-resolution tri-planes with 3D aware convolutions and an upsampler.
- StyleNeRF [6] generates a low-resolution latent features using NeRF and then upsample these features in 2D space with a StyleGAN-like structures.

None of these related methods are discussed and compared in this paper, making the judgement of "our approach is the first to propose a 3D-aware latent space" difficult. The suggestion would be add more discussion to clarify how the proposed approach differs from or improves upon these existing methods that used reduced resolution spaces for modeling NeRFs.

References:

[1] Mildenhall, Ben, et al. "Nerf: Representing scenes as neural radiance fields for view synthesis." Communications of the ACM 65.1 (2021): 99-106.

[2] Qian, Guocheng, et al. "Magic123: One Image to High-Quality 3D Object Generation Using Both 2D and 3D Diffusion Priors." The Twelfth International Conference on Learning Representations.

[3] Melas-Kyriazi, Luke, et al. "Realfusion: 360 reconstruction of any object from a single image. In 2023 IEEE." CVF Conference on Computer Vision and Pattern Recognition (CVPR). Vol. 1. 2023.

[4] Metzer, Gal, et al. "Latent-nerf for shape-guided generation of 3d shapes and textures." Proceedings of the IEEE/CVF Conference on Computer Vision and Pattern Recognition. 2023.

[5] Wang, Tengfei, et al. "Rodin: A generative model for sculpting 3d digital avatars using diffusion." Proceedings of the IEEE/CVF conference on computer vision and pattern recognition. 2023.

[6] Gu, Jiatao, et al. "Stylenerf: A style-based 3d-aware generator for high-resolution image synthesis." arXiv preprint arXiv:2110.08985 (2021).

**Questions:**

(1) In the first stage each latent tri-plane is trained individually, hence each tri-plane will have its own finetuned version of the image decoder. In section 4.2, when jointly training the IG-AE with the latent triplanes, which initialization is used for the decoder for IG-AE?

(2) Regarding the experiment setup: how many held-out scenes in each category (Cake, Figurine and House) for the Objeverse dataset? Does the same category appears in the training data of IG-AE ?

(2) In Table 5, it is shown that the InstantNGP exhibits both longer training time and rendering time paired with IG-AE (including the RGB decoding time for IG-AE). Given that the quality under IG-AE has no advantages, what are the advantages for the proposed IG-AE method compared with directly training NeRF using InstantNGP?

---

> ### Author Response · Authors · 2024-11-19
>
> We thank the reviewer for their constructive remarks. We address the reviewer's concerns about the chosen dataset in our [**global response**](https://openreview.net/forum?id=LTDtjrv02Y&noteId=mkPwBGuY4r). In this response, we provide specific answers to the other concerns and questions of the reviewer.
>
> ## (w.1) Justification of 3D-awareness
>
> ### (w.1.a.) Evaluation dataset
> We refer the reviewer to our [global response](https://openreview.net/forum?id=LTDtjrv02Y&noteId=mkPwBGuY4r), paragraph "Chosen datasets".
>
> ### (w.1.b.) Comparison with RGB NeRFs.
>
> We agree that latent NeRF quality still falls short of RGB NeRF. This is mainly due to the current limitation related to capturing high frequency details. However, latent NeRFs enable faster training times, faster rendering and inter-operability with other latent based 2D methods, while being compatible with different NeRF architectures. This is why we believe that this research direction is worth exploring. We refer to the [global response](https://openreview.net/forum?id=LTDtjrv02Y&noteId=mkPwBGuY4r) for a more detailed clarification of our motivation.
>
> ### (w.1.c.) Evaluation of 3D-awareness
>
> We thank the reviewer for the constructive remark regarding the evaluation of our 3D-aware latent space. We provide **animated videos** that showcase the view consistency of our latent NeRFs at the following link (https://anon-supp-iclr25.github.io/).
>
> Additionally, **their remarks have prompted us to include, in our upcoming paper revision**, quantitative NVS metrics computed in the latent space, which measure the ability of our latent NeRFs to replicate encoded latent images. We propose these metrics as a proxy for measuring the "3D-awareness" of our latent space.
>
> ### (w.1.d.) Choice of autoencoder
>
> We thank the reviewer for raising this interesting remark regarding the choice of our autoencoder. Indeed, a higher latent resolution might encourage local dependency over the autoencoder image and latent spaces, which may lead to more 3D-awareness than our base autoencoder. However, as they would still need some 3D-regularization in the latent space anyway, we have preferred to focus on low-resolution latent spaces as they provide a much greater advantage in terms of latent NeRF rendering/training times.
>
> In the current paper, we only consider the "Ostris KL-f8-d16" VAE as a base autoencoder. **To further explore this matter, we will include comparisons against other autoencoders** with different latent resolution and/or channel width in our upcoming paper revision.
>
> ## (w.2) Novelty over related work
> We thank the reviewer for bringing to light the suggested papers, which we will discuss in the upcoming revision. We would like to elaborate on some design choices regarding our work and the suggested related work:
>
> ### Low resolution NeRF and upsampling.
> - We agree that several methods learn compressed NeRF representation that are then upsampled either in a 3D space before rendering (Rodin) or in a 2D space  after rendering (StyleNeRF).
> - Our Latent NeRFs are also rendered in a similar fashion: a low-resolution rendering is upscaled. This rendering strategy is not novel.
>
> However, our work presents **key differences** that differentiate it from these works.
>
> - For Rodin, their encoder maps images to latent vectors that condition a generative model. These latent vectors are not latent images, and are specifically tailored for their pipeline. Hence, their latent space cannot be used as a proxy of the RGB space, and does not show compatibility with varied NeRF architectures.
> - StyleNeRF and other similar works warrant a separate discussion as they differ from our work in more technical aspects. We detail this below.
>
> ### Render v. Encode
> - Mainly, our work stands out from previous works because, once IG-AE is trained, (i) we are able to encode images into 3D-consistent features, (ii) this process does not require a NeRF. In a nutshell, our latent space can be used as a proxy of the real data for applications like novel view synthesis, unlike other works.
> - The suggested reference (StyleNeRF), and others (e.g. N3F [1]), cannot directly encode images into a latent space. They are only able to decode images from a latent space (e.g. for generation), in which case 3D-consistent features can only be obtained via a NeRF rendering. Handling an encoding scheme is the challenging part as the encoder outputs are not naturally 3D-consistent and need alignment.
> - As our work is the first to enable encoding data into 3D-aware latent representations, we claim to introduce the first 3D-aware latent space. To clarify this point, we propose to re-formulate our claim as: "**Our work is the first to propose an image autoencoder with a 3D-aware latent space**".

---

> ### Author Response · Authors · 2024-11-19
>
> ## Questions
>
> > 1. [...] In section 4.2, when jointly training the IG-AE with the latent triplanes, which initialization is used for the decoder for IG-AE?
>
> To train IG-AE, we have a first warmup stage in which Tri-Planes are trained while keeping the decoder frozen (this is done to avoid back-propagating random gradients into the autoencoder weights). As such, we begin the IG-AE training with the baseline unmodified AE modules (ref. lines 305-307).
>
> > 2. [...] how many held-out scenes in each category (Cake, Figurine and House) for the Objeverse dataset? Does the same category appears in the training data of IG-AE ?
>
> We use a category agnostic version of Objaverse throughout our paper. We simply hold-out scenes from this set to consistute our set of evaluation scenes.
>
> > 3. [...] Given that the quality under IG-AE has no advantages, what are the advantages for the proposed IG-AE method compared with directly training NeRF using InstantNGP?
>
> IG-AE indeed exhibits slower rendering+decoding/training times when compared against InstantNGP, while showcasing the lowest NVS performances across all the methods we tested. This illustrates that some current NeRF methods are more compatible with latent scene learning than others. We include InstantNGP in our comparisons to add further emphasis on the superiority of IG-AE over AE. One advantage in this case (as compared to RGB) would come from the inter-operability of a latent InstantNGP with latent-based 2D applications, which are not possible when using an InstantNGP model in the RGB space.
>
>
> ## References
>
> [1] Tschernezki, V., Laina, I., Larlus, D., & Vedaldi, A. (2022, September). Neural Feature Fusion Fields: 3D Distillation of Self-Supervised 2D Image Representations. 2022 International Conference on 3D Vision (3DV), 443–453.\

---

> > ### Comment · Reviewer_jLB8 · 2024-11-22
> >
> > > To train IG-AE, we have a first warmup stage in which Tri-Planes are trained while keeping the decoder frozen (this is done to avoid back-propagating random gradients into the autoencoder weights). As such, we begin the IG-AE training with the baseline unmodified AE modules (ref. lines 305-307).
> >
> > Thanks for the clarification:)
> >
> > >We use a category agnostic version of Objaverse throughout our paper. We simply hold-out scenes from this set to consistute our
> > set of evaluation scenes.
> >
> > Let me make my question clear: what **exactly** is the **number** of objects in your test dataset? I would expect a test set with enough variations to support claims in the paper.
> >
> > > IG-AE indeed exhibits slower rendering+decoding/training times when compared against InstantNGP, while showcasing the lowest NVS performances across all the methods we tested. This illustrates that some current NeRF methods are more compatible with latent scene learning than others. We include InstantNGP in our comparisons to add further emphasis on the superiority of IG-AE over AE. One advantage in this case (as compared to RGB) would come from the inter-operability of a latent InstantNGP with latent-based 2D applications, which are not possible when using an InstantNGP model in the RGB space.
> >
> > Some numbers for reference (all numbers is reported in this paper):
> > - I-NGP (RGB): 6 min training, 11.3 ms rendering, 37.78 PSNR
> > - IG-AE (highest-quality with Vanilla NeRF): 28 min training, 19.7 ms rendering, 33.60 PSNR
> > - IG-AE (fastest with I-NGP): 19 min training, 7.3 ms rendering, 27.47 PSNR
> >
> > I would say IG-AE has no advantages either in terms of quality and rendering/training times. The inter-operability might be an advantage; however (as I have mentioned in the previous comment), there are no experiment in the paper to support the inter-operability. At least there should be some showcases to demonstrate the inter-operability with latent-based 2D applications.

---

> ### Comment · Reviewer_jLB8 · 2024-11-22
>
> Thanks for the response. Here are my further comments:
> > In this work, we tackle the task of learning NeRFs in an AE latent space. As corroborated by our experimental results with a naive autoencoder, this task is particularly challenging even for simple object-level scenes. As such, we start by tackling synthetic objects, and defer from complex scenes, since even simple objects do not currently work well with latent NeRFs.
>
> I acknowledge that the task is challenging. Yet, my opinion on ShapeNet dataset remain unchanged - they are too simple in terms of appearances with simple diffuse textures, i.e., the conclusion verified on Shapenet dataset is difficult to generalize to more general scenarios.
> On the other hand, I agree that the Objaverse dataset is suitable for this task. However, I still have concerns that the proposed experiment setup on the Objaverse dataset is not convincing enough (see below).
>
> > These limitations exist concurrently, but the former is more prominent than the latter. Therefore, our work focuses on the former which we see as the primary obstacle against latent NeRFs. Our choice of synthetic evaluation scenes allows us to have metrics that are more correlated to the 3D-awareness of the latent space, which is our target, and less with the fact that high-frequencies are attenuated.
> > We agree that latent NeRF quality still falls short of RGB NeRF. This is mainly due to the current limitation related to capturing high frequency details.
>
> It is ok to focus on one specific aspect than another. The question, however, is that low-frequency results actually conceal the problem of "3D awareness" simply because an over-smoothed image will always look like "more 3D-consistent".
> An over-smoothed solution might solve the "prominent" problem of "3D awareness" in first glance, but also has less potential to fix the second problem of high-frequency attenuation.
>
> > However, latent NeRFs enable faster training times, faster rendering and inter-operability with other latent based 2D methods, while being compatible with different NeRF architectures. This is why we believe that this research direction is worth exploring. We refer to the global response for a more detailed clarification of our motivation.
>
> I would respectfully disagree with the claim of "compatible with different NeRF architecture". My concern is that the proposed method trade-off too much result fidelity for fast training and rendering time. The inter-operability with other latent based 2D methods seems to be a good point. However, there are no experiments to support such inter-operability is enabled with the proposed IG-AE.
>
> > We thank the reviewer for the constructive remark regarding the evaluation of our 3D-aware latent space. We provide animated videos that showcase the view consistency of our latent NeRFs at the following link (https://anon-supp-iclr25.github.io/).
>
> As the time of 11/22/2024 I found the link is broken. I would check again later to see if the link get fixed (or maybe I should change a network:))
>
> > their remarks have prompted us to include, in our upcoming paper revision, quantitative NVS metrics computed in the latent space
>
> I appreciate for that and looking forward to the results.
>
> >  we have preferred to focus on low-resolution latent spaces as they provide a much greater advantage in terms of latent NeRF rendering/training times.
>
> Again, a lower resolution space actually conceal the problem of 3D-awareness. I am very interested in the results of IG-AE in higher resolution latent spaces - does it also solves the problem of "3D awareness"?
>
> > To clarify this point, we propose to re-formulate our claim as: "Our work is the first to propose an image autoencoder with a 3D-aware latent space".
>
> I believe that is a good clarification.

---

> ### Comment · Reviewer_jLB8 · 2024-11-22
>
> Overall, I believe the response have clarified some of my questions. However, my main concerns (the low-quality issue, the advantage of IG-AE, and the lack of comprehensive evaluation) have not been addressed. I would remain my score for now but also open for further discussions.

---

> ### Author Response · Authors · 2024-11-22
>
> We thank the reviewer for their message. Until we submit the paper revision in a few days, we can already provide some further clarifications and additional results.
>
> ### Follow-up questions
>
> -  Regarding the animated videos of our scenes, the website (https://anon-supp-iclr25.github.io/) appears to be working from our side. In any case, we have updated our supplementary materials to include the results illustrated in the site (under `Rebuttal - supplementary videos/IG-AE Supplementary.html`).
>
> > what **exactly** is the **number** of objects in your test dataset?
> - For Objaverse, we have tested the adopted NeRF architectures on three objects (Table 4: Cake, House, and Figurine).
> - For Shapenet, the results on each of the 3 categories are averaged from 4 scenes in that category, for a total of 12 objects; cf. Table 1.
>
> > I would say IG-AE has no advantages either in terms of quality and rendering/training times
>
> We respectfully disagree. To draw this conclusion, the reviewer directly compares INGP with all other NeRF models. Instead, we highlight the **speed advantages of our Latent NeRFs for each NeRF model invidually** (22x, 4.5x, and 3x faster for Vanilla NeRF, TensoRF and K-Planes respectively). While our proposition does not currently bring speed advantages to INGP, it still does on all other adopted architectures.
>
> **Considering other NeRF architectures than INGP is still relevant.** Many recent works refrain from using INGP and adopt other methods instead, because they provide other benefits than state-of-the-art speed and performance:
> - HyperFields [1] and HyperDiffusion [2] use MLP parameterization, like Vanilla-NeRFs
> - CAD [3] and PI3D [4] use Tri-Planes (closely related to K-Planes), for their compatibility with 2D models.
> - Neural Point Cloud Diffusion [5] use Point NeRF for the disantanglement of shape and appearance.
>
> Therefore, our improvements for numerous other NeRF architectures have **clear potential usecases** in the research community.
>
> Finally, we think it is important to clarify the **positioning of our work**. The aim of our research is not to present a state-of-the-art method in terms of speed or quality. Our goal is to **explore a new direction/task** -- the possibility of training NeRF in an AE latent space across varied NeRF architecture. We believe that exploring such new research avenues, with motivated application perspectives, is valuable for the research community and crucial for future innovation.
>
> ### New results confirming 3D-awareness (w.1.c.)
>
> We are working hard towards delivering our additional results as soon as possible in the upcoming revision. For now, we can already share one set of the promised results which we will add to our paper.
>
> The following tables compile the **NVS performance of various NeRF methods within the latent space** of our IG-AE vs the standard AE latent space.
>
> | Objaverse (cake)     |         | **PSNR** | **SSIM** |
> |----------------------|---------|----------|----------|
> | **Vanilla-NeRF**     | AE      | 38.36    | 0.854    |
> |                      | IG-AE   | **50.31**| **0.989**|
> | **Instant-NGP**      | AE      | 35.87    | 0.768    |
> |                      | IG-AE   | **44.11**| **0.967**|
> | **TensoRF**          | AE      | 36.79    | 0.817    |
> |                      | IG-AE   | **47.37**| **0.983**|
> | **K-Planes**         | AE      | 36.85    | 0.798    |
> |                      | IG-AE   | **44.74**| **0.969**|
>
> | Shapenet (vase)      |         | **PSNR** | **SSIM** |
> |----------------------|---------|----------|----------|
> | **Vanilla-NeRF**     | AE      | 38.48    | 0.841    |
> |                      | IG-AE   | **48.20**| **0.979**|
> | **Instant-NGP**      | AE      | 35.94    | 0.734    |
> |                      | IG-AE   | **43.84**| **0.955**|
> | **TensoRF**          | AE      | 36.56    | 0.780    |
> |                      | IG-AE   | **46.34**| **0.967**|
> | **K-Planes**         | AE      | 36.53    | 0.755    |
> |                      | IG-AE   | **44.26**| **0.949**|
>
> As illustrated, NeRFs consistently show major performance improvements when learning a scene in our 3D-aware latent scene. Therefore, IG-AE consistently allows latent NeRFs to better fit the latent space than a standard AE, **confirming our claim that our IG-AE does exhibit significantly more 3D awareness than a standard AE**.

---

> > ### Comment · Reviewer_jLB8 · 2024-11-23
> >
> > Thanks for the further response.
> >
> > TL;DR: I still have some concerns after reading new responses and results.
> >
> > > Regarding the animated videos of our scenes, the website (https://anon-supp-iclr25.github.io/) appears to be working from our side. In any case, we have updated our supplementary materials to include the results illustrated in the site (under Rebuttal - supplementary videos/IG-AE Supplementary.html).
> >
> > The link works for me now and I have checked these results. From what I have observed:
> > - It looks like the IG-AE produces more view-consistent result for the vase case and slightly better for the hat case.
> > - For the vase case one can still observe some flickering at the inside region of the vase even for IG-AE - the low resolution of these results make it hard to conclude whether it is a view-inconsistent artifact or simply a bake-in highlight effect.
> > - All the three objects exhibits extremely simple color texture (i.e., almost a uniform color). The vase case has a slightly non-trivial shape (i.e., concave).
> > - IG-AE on I-NGP still have significantly 3D inconsistency.
> >
> > Overall, what I could conclude from these visualizations are:
> > - The view-consistency seems to be improved (to some extent) on the selected object on ShapeNet dataset.
> > - Low-resolution results smoothed out a lot of details.
> > - Concrete conclusions regarding more general cases still cannot be clearly drawn.
> >
> > > For Objaverse, we have tested the adopted NeRF architectures on three objects (Table 4: Cake, House, and Figurine).
> > For Shapenet, the results on each of the 3 categories are averaged from 4 scenes in that category, for a total of 12 objects; cf. Table 1.
> >
> > I have to say three objects is not convincing enough to support claims in the paper. I do understand that people have to subsample the objaverse dataset due to limited computational resources - but I believe three is too small. I believe at least double-digit number of objects (with some non-trival shape and textures) are required to draw some solid conclusions.
> >
> > > Finally, we think it is important to clarify the positioning of our work. The aim of our research is not to present a state-of-the-art method in terms of speed or quality.
> >
> > I do not require the proposed method to be "state-of-the-art" in terms of quality or time. What I do care about (and I hope I can get some insights) from this paper is that - what are some **unique** advatanges that the proposed IG-AE have?
> > As the paper claims (and I do acknowledge that), the IG-AE space has not been proposed before. However, that also means it is the duty of this paper to convince the field that the new proposed space is useful with advantanges. For example -
> > - NeRF have advantages because it greatly improves the reconstruction and the novel view synthesis quality.
> > - Gaussian-Splatting have advantages because it greatly reduces the rendering speed.
> > - VAE and VQ-VAE have advantages because it enables the scaling of generative models.
> > - Latents from the pre-trained Masked Autoencoder (or contrastive learning methods such as MoCo/BYOL) have advantages because it shows great performance on downstream understanding tasks (as demonstrated firstly by linear probe experiments and then a lot of follow-up finetuning works).
> >
> > That said, I have tried my best to explore what advatanges does the IG-AE have:
> > - Does it improves the 3D reconstruction quality? It seems not.
> > - Does it reduces the rendering time for novel view synthesis? Partly - but sacrifies too much for quality.
> > - Does it improves the performance on some other downstream tasks? Maybe - but no conclusions can be drawn from the paper.
> > - Does it supports some unique task that other latent spaces are struggling to handle? Maybe - but no conclusions can be drawn from the paper.

---

> > ### Comment · Reviewer_jLB8 · 2024-11-23
> >
> > >  Our goal is to explore a new direction/task -- the possibility of training NeRF in an AE latent space across varied NeRF architecture.
> >
> > I acknowledge that this paper firstly explored this new direction. However, that also means the paper should at least tell the field why this new direction have further potentials. This would be especially important given the current experiment results are rather preliminary; otherwise people would question the motivation of this paper - why should we train NeRFs in an AE space, given current results are not satisfactory?
> >
> > > Therefore, our improvements for numerous other NeRF architectures have clear potential usecases in the research community.
> >
> > > We believe that exploring such new research avenues, with motivated application perspectives, is valuable for the research community and crucial for future innovation.
> >
> > This is the third time I ask this question: **what are these motivated application perspectives and potential usecases**? The paper does not have any showcases to demonstrate the application. Even some concrete list of new applications (or old applications that can be improved) are now shown in the paper.
> >
> >
> > > The following tables compile the NVS performance of various NeRF methods within the latent space of our IG-AE vs the standard AE latent space.
> >
> > I acknowledge these new results which demonstrate that the IG-AE latent space is better aligned with the volumetric rendering of NeRF. However, I notice that while IG-AE outperforms the AE by a large margin in the latent space (8-12dB), the gap between the decoded RGB quality is significant narrowed (2-4dB). That would raise one concern that whether the gain is from the significant lower resolution, and whether the decoded quality can be maintained for higher resolution results.

---

> > > ### Author Response · Authors · 2024-11-25
> > >
> > > We thank the reviewer once again for their response.
> > >
> > > ### Application perspectives
> > >
> > > We start by further clarifying some points raised by the reviewer below.
> > >
> > > > What I do care about (and I hope I can get some insights) from this paper is that - what are some **unique** advatanges that the proposed IG-AE have?
> > >
> > > A unique advantage of IG-AE is that it **preserves 3D-consistency** in its latent space.
> > >
> > > > **what are these motivated application perspectives and potential usecases**? The paper does not have any showcases to demonstrate the application. Even some concrete list of new applications (or old applications that can be improved) are now shown in the paper.
> > >
> > > Generally, we agree with the reviewer saying that we did not explore ourselves any tasks other than Latent NeRF training. However, **the potential of IG-AE is demonstrated in our motivation with the support of several references** in the literature providing several usecases, including the following:
> > > - Latent-NeRF [1] perform scene generation by training latent NeRFs. Better quality could be achieved by training the latent NeRFs in the latent space of IG-AE, thanks to its 3D-aware nature.
> > > - ED-NERF [2] and Latent-Editor [3] perform scene editing. To do so, they use latent NeRF with some workarounds (adapters and refinement layers) to ensure their compatibility with the used VAE. With IG-AE, they would not need such workarounds, simplifying their pipeline and potentially improving their performances.
> > >
> > > Additionally, works like Marigold [4] perform depth estimation using an image AE. The 3D-awareness of IG-AE could be leveraged to produce more accurate depth maps.
> > >
> > > Our revised introduction aims at better exposing all this motivation.
> > >
> > > Overall, we have identified **concrete application perspectives** in the literature **for our IG-AE or future work in the research direction we have launched**. We believe this meets the standards of ICLR, leaving the evaluation of IG-AE performance in such specialized works outside the scope of our paper.
> > >
> > > ### References
> > >
> > > [1] Gal Metzer, Elad Richardson, Or Patashnik, Raja Giryes, and Daniel Cohen-Or. Latent-NeRF for Shape-Guided Generation of 3D Shapes and Textures. In Proceedings of the IEEE/CVF Conference on Computer Vision and Pattern Recognition (CVPR), pp. 12663–12673, June 2023.
> > >
> > > [2] Park, J., Kwon, G., & Ye, J. C. (2024). ED-NeRF: Efficient Text-Guided Editing of 3D Scene With Latent Space NeRF. The Twelfth International Conference on Learning Representations.
> > >
> > > [3] Khalid, U., Iqbal, H., Karim, N., Hua, J., & Chen, C. (2023). LatentEditor: Text Driven Local Editing of 3D Scenes. arXiv Preprint arXiv:2312. 09313.
> > >
> > > [4] Ke, B., Obukhov, A., Huang, S., Metzger, N., Daudt, R. C., & Schindler, K. (2024, June). Repurposing Diffusion-Based Image Generators for Monocular Depth Estimation. Proceedings of the IEEE/CVF Conference on Computer Vision and Pattern Recognition (CVPR), 9492–9502.

---

> > > > ### Comment · Reviewer_jLB8 · 2024-11-26
> > > >
> > > > > Overall, we have identified concrete application perspectives in the literature for our IG-AE or future work in the research direction we have launched. We believe this meets the standards of ICLR, leaving the evaluation of IG-AE performance in such specialized works outside the scope of our paper.
> > > >
> > > > Fine. At least there are some potential applications listed... I have no further comments.
> > > >
> > > > > We have revised our dataset section to better motivate our choice of dataset, and to illustrate why it is a suitable choice for the evaluation of latent NeRF architectures and the 3D-awareness of our latent space.
> > > >
> > > > The revised version claims "Such object-level datasets with limited fine details have become standard for large-scale 3D data, and are widely used in recent works (Liu et al.,2023;Shi et al.,2024; Liu et al.,2024)." I agree that object-level dataset has become standard for evaluation. However, I would still argue that this sentence is misleading and attempts to conceal the weak evaluation of this paper:
> > > >
> > > > - (1) Other works did not conduct evaluation with "object-level datasets with **limited** fine details", let alone “this has become a standard”. For the referenced related work, (Liu et al., 2023) used the Objaverse dataset. (Shi et al.,2024) used the Objaverse dataset. (Liu et al.,2024) used the Google Scanned Object dataset. The most related paper, Latent-NeRF[1], used the Objaverse dataset. All these method evaluate their method extensively on object datasets (for example, objaverse) that exhibits complicated shapes and textures. The IG-AE conducted main experiments on ShapeNet. ShapeNet is **definitely not** the standard for evaluate 3D reconstruction and NVS for NeRF related methods. The IG-AE only have three object results on the Objaverse dataset, and these three objects have simple shape and texture.
> > > >
> > > > - (2) Three objects on the Objaverse dataset cannot be considered as "large-scale". I do understand that evaluation IG-AE requires training one latent NeRF for each object, and it would be more time-consuming than single feed-forward methods. Yet, my expectation would be at least around 10 objects with enough variations (shape, texture, BRDF, etc.,) to evaluate the performance of IG-AE under different scenarios. This is a reasonable expectation, because the IG-AE is expected to be generalizable on various objects (otherwise its contribution would be significantly reduced). As a reference, the mentioned related papers all have a significant number of objects evaluated than IG-AE; the original NeRF paper (which requires much more training time then IG-AE) evaluated around 10 objects / scenes.
> > > >
> > > > For other part I believe there are no (at least no major) issues. The related works are well-discussed. The advantages over vanilla AE is evaluated.

---

> > > > > ### Author Response · Authors · 2024-11-26
> > > > >
> > > > > We would like to thank the reviewer for their further comments on our choice of datasets.
> > > > >
> > > > > First of all, we apologize for the previously revised dataset section. Indeed, we completely agree with the reviewer in that the papers mentioned use Objaverse and **not** Shapenet, and that the phrasing used was not accurate. We have submitted a new paper revision that updates the dataset section (in violet), and corrects the previous section. We hope that this revision resolves the reviewer's concerns, and remain open for further discussions.
> > > > >
> > > > > Additionally, to be clearer about the limitations, we have dedicated a specific section to discuss these aspects in our new revision.
> > > > >
> > > > > On another note, thanks to the extension of the discussion period, **we will be able to provide in a follow-up response, and in our website, more results on several additional Objaverse scenes**. We hope that this will address the reviewer's concern about the number of evaluated Objaverse objects. We look forward to discussing these results with the reviewer.

---

> ### Author Response · Authors · 2024-11-22
>
> ### References
>
> [1] Babu, S., Liu, R., Zhou, A., Maire, M., Shakhnarovich, G. &amp; Hanocka, R.. (2024). HyperFields: Towards Zero-Shot Generation of NeRFs from Text. <i>Proceedings of the 41st International Conference on Machine Learning</i>, in <i>Proceedings of Machine Learning Research</i> 235:2230-2247.
>
> [2] Erkoç, Z., Ma, F., Shan, Q., Nießner, M., & Dai, A. (2023, October). HyperDiffusion: Generating Implicit Neural Fields with Weight-Space Diffusion. Proceedings of the IEEE/CVF International Conference on Computer Vision (ICCV), 14300–14310.
>
> [3] Wan, Z., Paschalidou, D., Huang, I., Liu, H., Shen, B., Xiang, X., … Guibas, L. (2024, June). CAD: Photorealistic 3D Generation via Adversarial Distillation. Proceedings of the IEEE/CVF Conference on Computer Vision and Pattern Recognition (CVPR), 10194–10207.
>
> [4] Liu, Y.-T., Guo, Y.-C., Luo, G., Sun, H., Yin, W., & Zhang, S.-H. (2024, June). PI3D: Efficient Text-to-3D Generation with Pseudo-Image Diffusion. Proceedings of the IEEE/CVF Conference on Computer Vision and Pattern Recognition (CVPR), 19915–19924.
>
> [5] Schröppel, P., Wewer, C., Lenssen, J. E., Ilg, E., & Brox, T. (2024, June). Neural Point Cloud Diffusion for Disentangled 3D Shape and Appearance Generation. Proceedings of the IEEE/CVF Conference on Computer Vision and Pattern Recognition (CVPR), 8785–8794.

---

> ### Author Response · Authors · 2024-11-25
> **Paper Revision**
>
> As promised, we have submitted our paper revision with the changes discussed in the rebuttal, highlighted by the blue parts in our revision. We have provided a global message [here](https://openreview.net/forum?id=LTDtjrv02Y&noteId=Yjcz1Gp7aq) to highlight the changes and added experiments in our paper.
>
> Particularly, to address the reviewer's remarks:
>
> - **(w.1.a)** We have revised our dataset section to **better motivate our choice of dataset**, and to illustrate why it is a suitable choice for the evaluation of latent NeRF architectures and the 3D-awareness of our latent space.
> - **(w.2)** We have **added a new section to our related work** that contrasts our work from other works tackling feature fields or scene embeddings in the literature (including Rodin and StyleNeRF). Additionally, we have added a section and a summarizing table in the appendix, providing a more detailed discussion of recent works and highlighting their distinctions. Moreover, as discussed with the reviewer, we have reformulated our claim as "we propose IG-AE, the first image autoencoder embedding a 3D-aware latent space".
> - Moreover, **we have added additional experiments** to answer the reviewer's concerns and questions:
>     - **(w.1.c)** Table 8 illustrates the NVS metrics computed directly on latent images, **further demonstrating the 3D-awareness of our latent space**.
>     - **(w.1.d)** Table 4 **compares autoencoders with differents downscale factors for latent NeRF training**. In this set of experiments, our conclusion agrees with the reviewer in that a higher resolution latent space would bring better NVS performances (which might be due to a higher local dependency between the latent and RGB spaces). However, increasing the latent resolution comes at the cost of compromising the speed advantages we have showcased with latent NeRFs. This motivates our choice of an autoencoder with a good trade-off between NVS performance and training time. We have added this discussion in the paper (Appendix E).
>
> We remain open for further discussions.

---

> ### Comment · Reviewer_jLB8 · 2024-11-27
>
> Thanks for the response. Overall I have no other major issues except a more comprehensive evaluation. I am looking forward to those new results.

---

> > ### Author Response · Authors · 2024-11-28
> >
> > As promised, we have added a new batch of experiments on 15 new Objaverse objects, **for a total of 18 objaverse objects** in our **newly revised paper**.
> > **New videos** are anonymously available at https://anon-supp-iclr25.github.io/ .
> >
> > We hope this addresses the reviewer's concerns regarding varied shapes and textures by providing a diverse range of scenes.
> >
> > The additional experiments carry the same conclusions as before, further confirming that the IG-AE 3D-aware latent space is better suited than the one of a standard AE for latent NeRF learning.
> >
> > We refer the reviewer to our global answer [here](https://openreview.net/forum?id=LTDtjrv02Y&noteId=qTEY6C9PrR) "Supplementary Experiments", and remain open for further discussions.

---

> > > ### Comment · Reviewer_jLB8 · 2024-11-29
> > >
> > > Thanks for the response.
> > > I have checked these results - it looks not bad. I would raise my score accordingly.

---

### Official Review · Reviewer_RTXt · 2024-11-04

**Soundness:** 3
**Presentation:** 4
**Contribution:** 2
**Rating:** 6
**Confidence:** 4

**Summary:**

This paper introduces the concept of 3D-awareness into the latent space of autoencoders, through learning a latent NeRF and an Inverse Graphics AutoEncoder (IG-AE). The proposed latent NeRF is general and can be used as standard extension to different previous NeRF architectures, with implementation code in the supplementary. The proposed IG-AE jointly train latent NeRFs and standard AE to achieve 3D awareness in latent space, on both synthetic and real data. The experimental comparisons and analysis are mostly thorough and comprehensive, while still with some issues in justifying its motivations.

**Strengths:**

1. 3D-awareness of 2D image generation from autoencoders is an important issue, and this paper seems to be the first work to address it.
2. The proposed latent NeRF operates fully within the latent space, which is a standardized solution and can work as an open-source extension to established NeRF architectures. The authors submitted the code in the supplementary.
3. The training framework of latent NeRF and IG-AE is sensible, with detailed ablation study to justify the loss design.
4. The paper is very well-written and easy to follow, with method design and experiments to support its motivation and arguments.

**Weaknesses:**

Although I agree with the importance of 3D-awareness of 2D autoencoders and appreciate the authors' efforts, I still have some concerns/questions for the proposed method to address 3D-awareness with latent NeRF:
1. Is a latent NeRF really necessary? Does it bring more advantage or more damage to the standard 2D AEs, especially when the scenes are quite complicated? Learning a scene with a NeRF model tends to smooth out high-frequency details (easier to be 3D-inconsistent), which is also true for TV loss as discussed in Line 363. Is it possible that the latent NeRF learning and TV loss would force the encoder to remove high-frequency contents during joint training, which can not be recovered by the decoder? One example is the cake in Figure 6, IG-AE is over-smoothed comparing to RGB. Could you conduct quantitative evaluation to show the level of loss in high frequencies?
2. Additionally, if the intension is to bring 3D-awareness to 2D autoencoders, why not add NeRF constraints on the final RGB images instead of the latents? Maybe this would cause less information loss? Is it possible to compare the trade-off between using latent NeRF and NeRF in final RGB space?
3. On the condition of the lost high-frequency details, is this level of "3D-awareness" still helpful for the 2D autoencoders? The encoding and decoding reconstruction on natural images by 2D encoders seems to be fine for now, as shown in Figure 9 and other literatures. The problem emerges when it comes to novel image generation rather than simple reconstruction, which is untouched in this paper.
4. The experiments seem to be limited to simple objects? The Objaverse and Shapenet objects should be much simpler than the original domain of the pre-trained "Ostris KL-f8-d16" VAE. Is it possible to show the performance on more complicated scenes with rich high-frequency details?
5. If the focus is to to improve standard 2D autoencoders then the comparisons to AE should be the reconstruction of multiview images?
6. If the justification is novel view synthesis, the baseline methods should be other (latent) NeRF methods instead of AE? Since AE itself is not for this task. Table 5 does show some advantages in training and rendering time, but with significant sacrifice on NVS performance.

**Questions:**

1. Is there any convergence issue during IG-AE training? Do you simply jointly train all modules with all losses?
2. I haven't checked the computation of latent NeRF, so I'm not sure if there would be any problems.
3. Is this paper an improved version of the one in the supplementary? There seems to be quite some similarities.

---

> ### Author Response · Authors · 2024-11-19
>
> We thank the reviewer for their constructive remarks. We address the reviewer's concerns about the motivation of our paper as well as the chosen dataset in our [**global response**](https://openreview.net/forum?id=LTDtjrv02Y&noteId=mkPwBGuY4r). In this response, we provide specific answers to the other concerns and questions of the reviewer.
>
> ### (w.1) Necessity and impact of Latent NeRFs
> > Is a latent NeRF really necessary?
>
> Yes, **we absolutely need latent NeRFs** to obtain a 3D-aware latent space in which latent NeRFs can be trained. Otherwise, we do not have any 3D consistent latent images as learning signal for 3D-awareness. We also refer to the next point (w.2) on RGB supervision.
>
> > Does it bring more advantage or more damage [to autoencoders] ?
>
> As the reviewer pointed out, this process may damage the autoencoder, which is also evident in our ablation study (Table 2, "IG-AE (no Pr)"). To prevent this, we added the "AE Preservation" part of our pipeline, which aims to maintain the autoencoder performance on 2D tasks. As shown in Table 2 and Figure 9, our full IG-AE model performs well on the 2D reconstructive task even for high frequencies. High frequency drops only arise when using the AE for 3D tasks (learning latent NeRFs), and not on 2D tasks (reconstructing images). We will clarify this point in the revised paper.
>
> To complement this analysis, we will include in the revised version of the paper the evaluation of the AE 2D task (reconstruction) not only on *real images*, but also on *synthetic images*, where our IG-AE shows similar or better performances compared to a standard AE.
>
> ### (w.2) RGB Supervision
>
> If we understand correctly, the reviewer suggests to supervise the AE in the RGB space by using RGB NeRFs. In fact, **the supervision in the RGB space is essential and already present in our pipeline**. However, having RGB NeRFs is not needed for this supervision, as we already have ground truth 3D-consistent views in the RGB space, and do not need a NeRF to obtain them, hence sparing us the computational cost. This supervision is practically done via the loss $L_\mathrm{RGB}$.
>
> Alternatively, if the reviewer is referring to exclusively using $L_\mathrm{RGB}$ while ablating $L_\mathrm{latent}$, it is a reasonable alternative that we have already tried in our preliminary experiments. This yielded an IG-AE with worse performances in downstream latent scene learning.
>
> >  Maybe this would cause less information loss?
>
> Regarding additional information loss, it is not related to having (or not) supervision in the RGB space. It is a direct consequence of the enforcement of an additional constraint in the latent space. As discussed previously, this does not affect the AE performance on 2D reconstruction.
>
> ### (w.3) 3D-awareness and autoencoder performances for 2D tasks
>
> As previously discussed, high frequency drops only occur in the 3D task of latent scene learning. It should not impact neither positively nor negatively purely 2D tasks, like image reconstruction and image generation (Table 2 (IG-AE) and Figure 9).
>
> ### (w.4) Dataset choice
> We refer the reviewer to our [global response](https://openreview.net/forum?id=LTDtjrv02Y&noteId=mkPwBGuY4r), section "Chosen datasets".
>
> ### (w.5+w.6) Evaluations
>
> Our focus is to improve a 2D autoencoder when used for the 3D task of latent scene learning, while preserving its performances on 2D tasks. In our evaluation, we show that: (i) IG-AE and a regular AE have similar performance on the 2D task, (ii) IG-AE has superior performance than regular AE when used for downstream novel view synthesis.
>
> For latent scene learning, as discussed in our related work, the closest related work [1] do not utilize *purely* latent NeRFs, as it implements hybrid latent-RGB NeRFs. This makes them unsuitable for comparison. Our work is the first to propose a standardized Latent NeRF Training Pipeline compatible with different NeRF models. **As such, we have no direct competitor/baseline in the literature with the same setting**. This leaves us with two types of evaluations across NeRF models:
> - (i) comparison of latent scene learning using IG-AE and AE (naive baseline) which highlights the 3D-awareness of IG-AE,
> - (ii) comparison between latent scene learning and RGB scene learning which highlights the advantages in terms of compute, and the current limitations in terms of rendering quality.

---

> ### Author Response · Authors · 2024-11-19
>
> ### Questions
>
> > 1. Is there any convergence issue during IG-AE training? Do you simply jointly train all modules with all losses?
>
> We observed no training instabilities; it was straightforward to find a suitable learning rate. We train all modules with all losses simultaneously.
>
> > I haven't checked the computation of latent NeRF, so I'm not sure if there would be any problems.
>
> We are not sure we understand this remark. What kind of problem is the reviewer referring to?
>
> > 3. Is this paper an improved version of the one in the supplementary? There seems to be quite some similarities.
>
> The paper in the supplementary is a separate contribution. It solves a different problem found in the literature: jointly learning numerous semantically similar scenes with reduced costs.
>
> To accomplish this, the authors propose a Micro-Macro Tri-Plane decomposition that works better when used in the latent space, hence adapting the proposed IG-AE training scheme from our paper to their targeted problem.
>
> ### References
>
> [1] Aumentado-Armstrong, T., Mirzaei, A., Brubaker, M. A., Kelly, J., Levinshtein, A., Derpanis, K. G., & Gilitschenski, I. (2023). Reconstructive Latent-Space Neural Radiance Fields for Efficient 3D Scene Representations. arXiv Preprint arXiv:2310. 17880.\

---

> ### Author Response · Authors · 2024-11-25
> **Paper Revision**
>
> As promised, we have submitted our paper revision with the changes discussed in the rebuttal, highlighted by the blue parts in our revision. We have provided a global message [here](https://openreview.net/forum?id=LTDtjrv02Y&noteId=Yjcz1Gp7aq) to highlight the changes and added experiments in our paper.
>
> Particularly, to address the reviewer's remarks:
>
> - **(w.1)** We have revised the introduction to **better illustrate the motivation** of our work, as well as the advantages and applications latent NeRFs bring.
> - **(w.4)** We have revised our dataset section to **better motivate our choice of dataset**, and to illustrate why it is a suitable choice for the evaluation of latent NeRF architectures and the 3D-awareness of our latent space.
> - We also added an experiment (Table 9) that showcases autoencoding metrics on synthetic images. As the table illustrates, our autoencoder showcases better PSNR and SSIM metrics on synthetic images, and slightly worse LPIPS metrics.
>
> We hope that our revision answers the reviewer's questions and concerns.

---

> ### Comment · Reviewer_RTXt · 2024-11-26
>
> I do appreciate the authors' efforts in addressing questions from all reviewers. After reading all the responses, I still share some of the concerns with Reviewer jLB8.
>
> The key issue from my opinion is the paper seems to be half-done on its way and might leave the audiences confused. An image autoencoder with a 3D-aware latent space is indeed intriguing, but what does it lead to? Better reconstruction or better generation? Or faster in speed but with comparable quality? Or maybe none of those? This is actually why I asked about the "necessity", "focus" and "justification" in the initial review. Comparing to a standard AE (such as Table 1 and other main results) is more like comparing to a naive baseline than justifying the necessity of the proposed pipeline for "concrete purposes". I believe a set of down-stream experiments would make the significance of the proposed 3D-aware latent space much more convincing.
>
> Again, I acknowledge that this paper proposes a valuable attempt to address general 3D-aware latent space for autoencoders, but whether the current experiments and analysis are enough to justify its significance for now remains questionable to me.
>
> If this paper should be accepted in its current version, at least the authors could be more frontal about its limitations comparing to various other pipelines for NVS (latent or not, such as in new Table 7) to help the audiences better understand its position in a bigger picture, rather than only emphasizing its advantages over the standard AE.

---

> > ### Author Response · Authors · 2024-11-26
> >
> > We would like to thank the reviewer for their response.
> >
> > > I believe a set of down-stream experiments would make the significance of the proposed 3D-aware latent space much more convincing.
> >
> > We agree with the reviewer but would like to point out that **the difficulty and the novelty of the task, the demonstrated efficiency gains, and the demonstrated performance gains compared to the baseline**, make our contributions already original and significant for a single paper. We refer to our new general response [here](https://openreview.net/forum?id=LTDtjrv02Y&noteId=cCj6vvrVMJ) for a more elaborate discussion.
> >
> > > at least the authors could be more frontal about its limitations comparing to various other pipelines for NVS (latent or not, such as in new Table 7) to help the audiences better understand its position in a bigger picture, rather than only emphasizing its advantages over the standard AE.
> >
> > We thank the reviewer for this suggestion, which we take into account in a new paper revision as follows.
> >
> > To ensure clarity and avoid potential misinterpretation of our results on latent NeRF learning, we have relocated the discussion of our work's limitations, previously embedded within the text, to a **dedicated limitations section** (section 5.4 in the paper).
> > This new section explicitly tackles the following limitations:
> >
> > - Low NVS performance compared to RGB,
> > - Limitation of Latent NeRFs in reproducing high frequency details in the RGB space.
> >
> > We hope that this addresses the reviewer's concern and remain open for further discussions.

---

### Author Response · Authors · 2024-11-19
**General response**

We thank the reviewers for dedicating their time and efforts to provide useful and relevant feedback about our work. We especially value that:
- generally, the reviewers have recognized the importance of the issue of 3D-awareness of 2D image autoencoder raised in this paper (RTXt s.1, jLB8 s.1) and the novel use of inverse graphics in the latent space (RTXt s.1, iRnM s.1);
- technically, they have appreciated our training framework (jLB8 s.2) and our detailed ablation study (RTXt s.3), as well as our open-source extension of Nerfstudio (RTXt s.2, jLB8 s.4, iRnM s.4);
- structurally, they have generally understood the paper (jLB8 s.2) and found it "very well written and easy to follow, with method design and experiments to support its motivation and arguments" (RTXt s.4).

We provided an individual response to each reviewer, and additionally address common concerns below. We will provide the revised version of the paper as soon as possible, as to facilitate post-revision discussions. Supplementary videos can be (anonymously) found at https://anon-supp-iclr25.github.io. We look forward to discussing with the reviewers.

---

> ### Author Response · Authors · 2024-11-19
>
> ## Paper motivation
>
> The reviewers helped us identify a shortfall in the presentation of the paper's motivation (RTXt w.1, iRnM w.2), particularly in the introduction. We acknowledge this issue and will revise the introduction to enhance clarity, as it directly impacts both the perceived importance of the work and the expectations of the reader. In brief, our motivation for this work is as follows.
> - Latent image representations are **standard in machine learning** and have proved to be advantageous for 2D computer vision tasks [1,2], where pretrained autoencoders have been leveraged to operate in the latent space instead of the image space for efficiency purposes.
> - Latent image representations have not yet been properly adopted for Inverse Graphics, even though they could bring major advantages.
>    - *Similarly to 2D computer vision*, applying NeRFs on lower-resolution latent images enables **significant speed-ups** in both rendering and training times. While NeRF speed improvements have traditionally focused on working directly on NeRF techniques, this work's research direction works instead on the space in which NeRF models are trained. This means that this research direction offers a **broader potential impact** across various NeRF models, extending beyond a single specific NeRF improvement, as observed in Table 5 of the paper.
>     - *Furthermore*, latent NeRFs unlock an **inter-operability** between NeRFs (3D) and pretrained latent-based methods (2D), using the latter for applications like scene generation [3] or scene editing [4]. However, the **3D inconsistency of the 2D latent space has been a limiting factor** throughout these works, hence requiring special adapters to properly integrate NeRFs in the latent space.
> - The points above suggest the need for a __common, universal, AE latent space__ that is tailored to NeRFs by adding 3D-awareness/underlying geometry. Our work represents a first effort towards this potential foundation model.
>
> Our current introduction only hints at the presented points. We will thoroughly revise it to make sure all points clearly presented and that the paper is better positioned in the current literature context.
>
>
> ## Choice of datasets
>
> In our experiments, we evaluate our IG-AE by evaluating it:
> - with varied NeRF architectures,
> - on different synthetic datasets: Objaverse (held-out) and Shapenet (out-of-distribution).
>
> Thanks to the feedback of the reviewers, we have realized that we should further motivate our choice of datasets. We will integrate this discussion in the revised paper.
>
> The reviewers (RTXt w.4, jLB8 w.1a) found the datasets on which we evaluate our 3D-aware latent space to be overly limited to simple object-level scenes. Below, we explain why the chosen datasets are suited for our evaluation and adequate to support our claims.
>
> ### Getting the basics right: synthetic scenes are challenging
>
> In this work, we tackle the task of learning NeRFs in an AE latent space. As corroborated by our experimental results with a naive autoencoder, **this task is particularly challenging even for simple object-level scenes**. As such, we start by tackling synthetic objects, and defer from complex scenes, since even simple objects do not currently work well with latent NeRFs.
>
> The difficulty lies in two key limiting factors identified in our paper:
> - lack of 3D-awareness of the latent space,
> - inability for latent NeRFs to capture high-frequency details.
>
> These limitations exist concurrently, but the former is more prominent than the latter. Therefore, our work focuses on the former which we see as the primary obstacle against latent NeRFs. Our choice of synthetic evaluation scenes allows us to have metrics that are more correlated to the 3D-awareness of the latent space, which is our target, and less with the fact that high-frequencies are attenuated.
>
> ### Aligning evaluation data with training data
>
> Training IG-AE requires a large dataset of varied 3D scenes with precise camera annotations. **This is only available at the time of writing as synthetic object-level datasets.** Therefore, evaluating IG-AE on synthetic data is appropriate, as it aligns with the training domain.
>
> A broader consequence of this is that synthetic, object-level datasets have become standardly adopted when large-scale 3D data is necessary. Among many others, [5, 6, 7] all use synthetic data to train and evaluate their methods.
> In this regard, our work already **has direct application perspectives in such works**.

---

> ### Author Response · Authors · 2024-11-19
>
> ## References
>
> [1] Robin Rombach, Andreas Blattmann, Dominik Lorenz, Patrick Esser, and Björn Ommer. High-Resolution Image Synthesis With Latent Diffusion Models. In Proceedings of the IEEE/CVF Conference on Computer Vision and Pattern Recognition (CVPR), pp. 10684–10695, June 2022.\
> [2] Patrick Esser, Robin Rombach, and Bjorn Ommer. Taming Transformers for High-Resolution Image Synthesis. In Proceedings of the IEEE/CVF Conference on Computer Vision and Pattern Recognition (CVPR), pp. 12873–12883, June 2021.\
> [3] JangHo Park, Gihyun Kwon, and Jong Chul Ye. ED-NeRF: Efficient Text-Guided Editing of 3D Scene With Latent Space NeRF. In The Twelfth International Conference on Learning Representations, 2024.\
> [4] Umar Khalid, Hasan Iqbal, Nazmul Karim, Jing Hua, and Chen Chen. LatentEditor: Text Driven Local Editing of 3D Scenes. arXiv preprint arXiv:2312.09313, 2023.\
> [5] Liu, R., Wu, R., Van Hoorick, B., Tokmakov, P., Zakharov, S., & Vondrick, C. (2023). Zero-1-to-3: Zero-shot One Image to 3D Object. 2023 IEEE/CVF International Conference on Computer Vision (ICCV), 9264–9275.\
> [6] Shi, Y., Wang, P., Ye, J., Mai, L., Li, K., & Yang, X. (2024). MVDream: Multi-view Diffusion for 3D Generation. The Twelfth International Conference on Learning Representations.\
> [7] Liu, Y., Lin, C., Zeng, Z., Long, X., Liu, L., Komura, T., & Wang, W. (2024). SyncDreamer: Generating Multiview-consistent Images from a Single-view Image. The Twelfth International Conference on Learning Representations.\

---

### Author Response · Authors · 2024-11-25
**Paper Revision**

We thank the reviewers once again for the valuable reviews and discussions. As promised, we have submitted our paper revision with the changes discussed in the rebuttal, highlighted in blue in our revision.
- *(Reviewers RTXt (w.1), iRnM (w.2), jLB8)* We have **revised the introduction to better illustrate the motivation** of our work, as well as the advantages and applications brought by latent NeRFs.
- *(Reviewers jLB8 (w.2), iRnM (w.3))* We have **added a new section to our related work** that contrasts our work from other works tackling feature fields or scene embeddings in the literature, including the suggested references. Additionally, we have added a section and a summarizing table in the appendix (section A, table 3), providing a more detailed discussion of recent works and highlighting their distinctions.
- *(Reviewers RTXt (w.4), jLB8 (w.1.a))* We have revised our dataset section (5.2) to **better motivate our choice of dataset**, and to illustrate why it is a suitable choice for the evaluation of latent NeRF architectures and the 3D-awareness of our latent space.
- Moreover, **we have added additional experiments** to answer the reviewer's concerns and questions:
    - *(Reviewer jLB8 (w.1.c))* Table 8 (5.3, table in appendix D.2) reports NVS metrics computed directly on latent images, **further demonstrating the 3D-awareness of our latent space**.
    - *(Reviewer jLB8 (w.1.d))* Table 4 (appendix E) justifies our choice of autoencoder by **evaluating other autoencoders** on our task of latent NeRF training. This set of experiments further explains our choice of autoencoder, which is a compromise between NVS performance and training/rendering speeds.
    - *(Reviewer RTXt (w.2))* Table 9 (appendix D.2) reports **autoencoding performance on synthetic images**.
- Additionally, we have already provided the reviewers with supplementary illustrations available at https://anon-supp-iclr25.github.io/ that showcase animated videos highlighting the 3D-awareness of our latent space and the frame-to-frame consistency of our NeRFs.

We remain available for further discussions.

---

### Author Response · Authors · 2024-11-26
**Regarding the paper's scope and downstream applications**

We would like to thank the reviewers for the animated and constructive discussion, which we believe has greatly helped us to improve our paper and clarified most of the raised concerns. One remaining concern shared by Reviewers RTXt and jLB8 is the lack of experiments on downstream applications. We would like to clarify our position on this matter in a separate thread for the sake of clarity.

Firstly, we reiterate and summarize our paper's contributions as acknowledged, to our understanding, by all reviewers.
- We identify a **new research direction, 3D-aware image latent spaces**, with **application perspectives clearly grounded in the literature**:
    1. efficiency gains by manipulating latent instead of RGB NeRFs for 3D tasks, where performance is hindered by 3D-inconsistent latent spaces;
    2. interoperability between NeRFs (3D) and pretrained latent-based methods (2D) which, until now, has necessitated special adapters due to 3D-inconsistent latent spaces.
- To tackle this challenging task -- as the only baseline, a naive autoencoder, does not work well even on simple cenes --, we propose **the first 3D-aware autoencoder**.
- We experimentally **confirm for the task of novel view synthesis**:
    - the 3D awareness of our autoencoder and its positive consequence on performance;
    - the efficiency gains of latent NeRFs.

We have debated with the reviewers over the following question:
> **Should the paper include experiments on downstream applications referenced in the motivation to show its advantages?**

**We already show the mentioned speed advantages (perspective 1)** (discussed in Sections 5.2 and D.2). We acknowledge that we do not include experiments in downstream tasks (perspective 2) and that, despite our performance gains relative to the current AE baseline, we do not match the performance of the RGB methods.

Still, given the **difficulty and novelty of the task** and the already demonstrated benefits, we think our submission can be considered as **a first step in this direction**, leaving other downstream applications outside our scope.

---

### Author Response · Authors · 2024-11-28
**Supplementary Experiments**

This is to inform the reviewers that we have added **experiments on 15 additional Objaverse objects** in a newly revised version of our paper, as suggested by reviewer jLB8. These experiments compare IG-AE and AE across the four NeRF representations for each object, **which adds 60 new comparisons**.

The quantitative results for these new experiments can be found in **tables 5 to 9** in the appendix (page 18, in brown).
For the qualitative results, we have **added new animated videos on our anonymous website** (https://anon-supp-iclr25.github.io/), and will be adding new comparative figures with visuals in the camera-ready version of our paper.

These results further support our previous claims of having a 3D-aware latent space better suited for latent NeRFs.

We remain open for further discussions.

---

### Meta-Review · Area_Chair_YwPM · 2024-12-14

**Metareview:**

The authors present a method to align a 2D image autoencoder with a 3D latent space. The authors demonstrate that the resulting latent space leads to better NVS quality with less 2D artifacts.

The reviewers converged to positively-leaning scores of 6/6/6 over the course of discussion, appreciating the interesting new idea and novelty of this work and acknowledging the clear shown benefit. At the same time, questions regarding relevancy remain.

I decided to follow the reviewers recommendation and propose to accept the paper. I think this is an interesting paper that can spark some novel ideas.

**Additional Comments On Reviewer Discussion:**

The paper was intensively discussed in the author-reviewer discussion phase, leading to significant improvements of the paper over the course of discussion. In the end, all reviewers could be convinced to vote for acceptance.

---

### Decision · Program_Chairs · 2025-01-22

Accept (Poster)